# DISCRETE GUIDANCE MATCHING: EXACT GUIDANCE FOR DISCRETE FLOW MATCHING

**Zhengyan Wan**[*1]   **Yidong Ouyang**[*2]   **Liyan Xie**[3]   **Fang Fang**[†1]   **Hongyuan Zha**[†4]

**Guang Cheng**[2]

[*] **Equal contribution**   [†] **Corresponding author**
[1] School of Statistics, East China Normal University
[2] Department of Statistics, University of California, Los Angeles
[3] Department of Industrial and Systems Engineering, University of Minnesota
[4] School of Data Science, The Chinese University of Hong Kong, Shenzhen

## ABSTRACT

Guidance provides a simple and effective framework for posterior sampling by steering the generation process towards the desired distribution. When modeling discrete data, existing approaches mostly focus on guidance with the first-order approximation to improve the sampling efficiency. However, such an approximation is inappropriate in discrete state spaces since the approximation error could be large. A novel guidance framework for discrete data is proposed to address this problem: we derive the exact transition rate for the desired distribution given a learned discrete flow matching model, leading to guidance that only requires a single forward pass in each sampling step, significantly improving efficiency. This unified novel framework is general enough, encompassing existing guidance methods as special cases, and it can also be seamlessly applied to the masked diffusion model. We demonstrate the effectiveness of our proposed guidance on energy-guided simulations and preference alignment on text-to-image generation and multimodal understanding tasks. The code is available at `https://github.com/WanZhengyan/Discrete-Guidance-Matching`.

## 1 INTRODUCTION

Discrete diffusion models (Austin et al., 2021; Campbell et al., 2022; Sun et al., 2023; Lou et al., 2024) and discrete flow-based models (Campbell et al., 2024; Gat et al., 2024; Shaul et al., 2025; Qin et al., 2025) have received significant attention for generating samples in discrete state spaces, providing effective alternatives to autoregressive (AR) models. Additionally, several works aim to study how to guide a pre-trained discrete model towards a desired distribution (Vignac et al., 2023; Schiff et al., 2025; Nisonoff et al., 2025). However, there are some challenges to developing guidance mechanisms in discrete cases. The main challenge is that discrete guidance is often associated with a transition probability or transition rate, which requires extra computation for all possible target positions after the transition. To reduce the number of forward passes through guidance models, existing methods treat the model as a continuous function and use the first-order approximation to compute guidance efficiently (Vignac et al., 2023; Schiff et al., 2025; Nisonoff et al., 2025). Unfortunately, this approximation might introduce non-negligible approximation errors. Moreover, existing literature only consider some particular cases, like class conditional guidance or energy-weighted guidance, which might not be general enough for various tasks.

We seek to develop a general guidance framework for discrete diffusion and flow-based models, which is illustrated in Fig. 1. Leveraging the Continuous-Time Markov Chain (CTMC) framework, given a pretrained model for sampling from a source distribution and the density ratio between the target distribution and source distribution, our method identifies the exact transition rate for sampling the target distribution. To the best of our knowledge, our guidance framework is the *first* exact discrete guidance in general form. Since our guidance formulation is expressed by the density

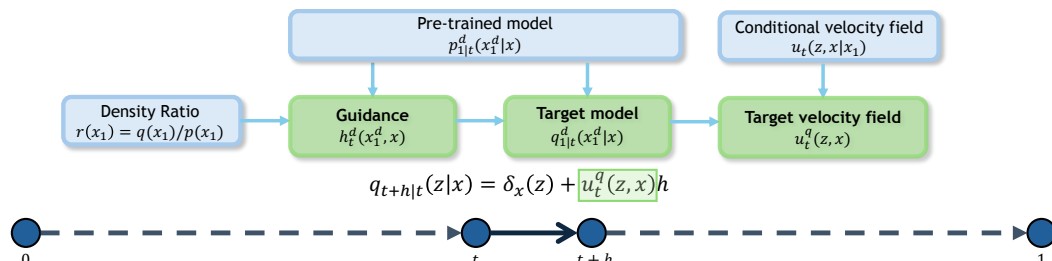

Figure 1: Framework of the proposed guidance on discrete flow matching. Given a pretrained discrete flow matching on source distribution $p_1(x)$ and a known density ratio $r(x) = q_1(x)/p_1(x)$, our framework calculates the target velocity field to generate target distribution $q_1(x)$. Our framework is general enough to deal with energy-guided sampling with $p_1^{(\gamma)}(x) \propto p_1(x)e^{-\gamma \mathcal{E}(x)}$ shown in Section 4.1, classifier guidance shown in Theorem 1, and preference alignment in Section 4.2.

Table 1: Overall comparison with existing discrete guidance. Our posterior-based guidance provides an exact formulation of guidance for general target distributions with a sampling-efficient implementation. Existing methods are either limited to class-conditional generation or suffer from non-negligible approximation errors with multiple function evaluations in each sampling step. A more detailed discussion is provided in Section 3.

| Guidance Method | Formula | Exact Guidance | # of function evaluations |
|---|---|---|---|
| Posterior-Based (Ours) | Equation 6 | ✓ | 1 |
| Rate-Based Nisonoff et al. (2025) | Equation 9 | ✓ | $\mathcal{D} + 1 \,;\, \mathcal{D} \times (|\mathcal{S}| - 1) + 1$ |
| First-Order Approximated Nisonoff et al. (2025) Vignac et al. (2023); Schiff et al. (2025) | Equation 13 | ✗ | 2 |

*Note:* In Nisonoff et al. (2025), guidance requires $\mathcal{D} \times (|\mathcal{S}| - 1) + 1$ function evaluations. When implemented using Algorithm 3, the number of function evaluations reduces to $\mathcal{D} + 1$.

ratio between the target distribution and source distribution, it also encompasses existing guidance methods as special cases. This is because the density ratio can be calculated by the energy function in energy-guided sampling, the ratio of the classifier in class-conditional generation, and preference alignment in reinforcement learning from human feedback (RLHF). In Table 1, we compare the formulation of the guidance and the number of function evaluations in each sampling step. Our framework only requires one forward pass in each sampling step for any initial distribution.

To compute the proposed guidance, we employ the Bregman divergence to train a network for estimating the conditional expectation. To further utilize potentially available samples from the target distribution, we further propose a regularization term.

In summary, our main contributions are as follows.

1. We introduce a general guidance framework for discrete flow matching, achieving efficient sampling without approximation.

2. We propose to learn the guidance network by minimizing Bregman divergence. We further propose a regularization technique to utilize samples from the target distribution.

3. We verify the effectiveness of the proposed framework on energy-based sampling through simulations, preference alignment on text-to-image, and multimodal understanding benchmark.

NOTATION. Let $[N] = \{1, 2, \ldots, N\}$ for a positive integer $N$. We use $\dot{\kappa}_t$ to denote the time derivative of a function $\kappa_t$ with respect to $t$. For a $\mathcal{D}$-dimensional vector $z$, let $z^d$ and $z^{\backslash d}$ denote the $d$-th element of the vector $z$ and the $(\mathcal{D}-1)$-dimensional vector $(z^1, \ldots, z^{d-1}, z^{d+1}, \ldots, z^{\mathcal{D}})^\top$. For a positive integer $p$, let $\mathbb{1}_p$ denote $p$-dimensional all-one vector. We write $f(h) = o(h)$ if $f : \mathbb{R} \to \mathbb{R}$ satisfies $f(h)/h \to 0$ as $h \to 0^+$. For two quantities $x, z$, define the Kronecker delta $\delta_x(z)$ satisfying $\delta_x(z) = 1$ if $x = z$ and $\delta_x(z) = 0$ if $x \neq z$. Let $\langle x, z \rangle$ be the inner product of two vectors $x, z$ in Euclidean space. We denote $\|x\|_2$ as the Euclidean norm of a vector $x$.

## 2 DISCRETE FLOW MODELS

Discrete flow models (DFMs) aim to generate samples from a $\mathcal{D}$-dimensional discrete state-space with probability mass function (pmf) $q_1$, defined on the domain $\mathcal{S}^{\mathcal{D}}$, where $\mathcal{S}$ denotes the finite state space of each dimension. A natural example of this setting is text generation, where each coordinate corresponds to a token drawn from the vocabulary $\mathcal{S}$. To begin with, we first introduce some background on discrete flow models (Campbell et al., 2024; Gat et al., 2024; Shaul et al., 2025) under the framework of CTMC (Norris, 1998), which we describe in detail in Appendix C.

### 2.1 TRANSITION RATE AND TRANSITION PROBABILITY

Given a CTMC with the marginal pmfs $\{q_t\}_{t \in [0,1]}$, let $u_t^q$ be the associated transition rate of this CTMC at time $t$, which is a $|\mathcal{S}|^{\mathcal{D}} \times |\mathcal{S}|^{\mathcal{D}}$ matrix. For $z, x \in \mathcal{S}^{\mathcal{D}}$ with $z \neq x$, we write $u_t^q(z, x)$ for the corresponding entry in this matrix representing the intensity of the transition from state $x$ to state $z$ at time $t$. If $u_t^q(z, x)$ satisfy the following rate-properties:

$$u_t^q(z, x) \geq 0, \text{ for any } z \neq x, \text{ and } \sum_{z \in \mathcal{S}^{\mathcal{D}}} u_t^q(z, x) = 0,$$

and the Kolmogorov forward equation (also known as the continuity equation):

$$\dot{q}_t(x) = \sum_{z \in \mathcal{S}^{\mathcal{D}}} u_t^q(x, z) q_t(z) = \underbrace{\sum_{z \neq x} u_t^q(x, z) q_t(z)}_{\text{incoming flux}} - \underbrace{\sum_{z \neq x} u_t^q(z, x) q_t(x)}_{\text{outgoing flux}}, \tag{1}$$

we say $u_t^q$ can generate the probability path $q_t$. Based on the above Kolmogorov forward equation, for a sufficiently small time step $h$, we have

$$q_{t+h}(x) = q_t(x) + \dot{q}_t(x) h + o(h) = \sum_{z \in \mathcal{S}^{\mathcal{D}}} \left\{ \delta_z(x) + u_t^q(x, z) h + o(h) \right\} q_t(z),$$

which implies that the transition probability from state $z$ to state $x$, for any $x, z \in \mathcal{S}$, is

$$q_{t+h|t}(x|z) = \delta_z(x) + u_t^q(x, z) h + o(h). \tag{2}$$

Thus, the transition rate $u_t^q$ is also the generator of the CTMC $\{q_t\}_{t \in [0,1]}$ (Holderrieth et al., 2025).

### 2.2 MODELING TRANSITION RATE VIA MARGINALIZATION TRICK

Given the transition rate $u_t^q(z, x)$ and current state $x$ at time $t$, we can generate a sample from $q_{t+h}$ using the equation 2, with a sufficiently small time step $h$. Therefore, we can learn a transition rate $u_t^q$ that can transport from an initial noise distribution $q_0$ to a target data distribution $q_1$. To obtain such a transition rate for sampling, a natural method is to learn the conditional expectation of the conditional transition rate $u_t^q(z, x|\mathbf{x}_1)$ that generates the conditional probability path $q_{t|1}(\cdot|\mathbf{x}_1)$ interpolating from noise to the datapoint $\mathbf{x}_1$. Then $u_t^q(z, x) = \mathbb{E}_{q_{1|t}(\mathbf{x}_1|x)}[u_t^q(z, x|\mathbf{x}_1)]$ can generate the target probability path $q_t$ (see Proposition 3.1 of Campbell et al., 2024), i.e., it satisfies the Kolmogorov forward equation 1. For completeness, we include the proof in Appendix D.1. We are free to define the conditional probability path $q_{t|1}(\cdot|\mathbf{x}_1)$ and with the corresponding conditional transition rate as needed.

It is worth noting that learning a $|\mathcal{S}|^{\mathcal{D}}$-dimensional vector-valued function $(u_t(z, x))_{z \in \mathcal{S}^{\mathcal{D}}}$ of current time $t$ and state $x$ is intractable when $\mathcal{D}$ is relatively large. To handle such a high-dimensional setting, a common approach is to construct a coordinate-wise conditional probability path and transition rate,

$$q_{t|1}(x|\mathbf{x}_1) = \prod_{d=1}^{\mathcal{D}} q_{t|1}^d(x^d|x_1^d), \text{ and } u_t^q(z, x|\mathbf{x}_1) = \sum_{d=1}^{\mathcal{D}} \delta_{x^{\backslash d}}(z^{\backslash d}) u_t^{q,d}(z^d, x^d|x_1^d), \tag{3}$$

which means that the elements of the vector $\mathbf{x}_t$ are independent conditional on $\mathbf{x}_1$. Here, $u_t^{q,d}(z^d, x^d|x_1^d)$ is the conditional transition rate that generates the conditional probability path $q_{t|1}^d$.

A popular choice of probability path and the associated conditional transition rate used in the previous works (Campbell et al., 2024; Gat et al., 2024) is

$$q_{t|1}^d(x^d|x_1^d) = (1 - \kappa_t)q_0^d(x^d) + \kappa_t \delta_{x_1^d}(x^d); \; u_t^{q,d}(z^d, x^d|x_1^d) = \frac{\dot{\kappa}_t}{1 - \kappa_t}(\delta_{x_1^d}(z^d) - \delta_{x^d}(z^d)), \quad (4)$$

where $\kappa_t : [0,1] \to [0,1]$ is a non-decreasing function satisfying $\kappa_0 = 0$ and $\kappa_1 = 1$.

After defining the conditional path and rate, the marginal transition rate is given by

$$u_t^q(z, x) = \sum_{\mathbf{x}_1} u_t^q(z, x|\mathbf{x}_1)q_{1|t}(\mathbf{x}_1|x) = \sum_{d=1}^{\mathcal{D}} \delta_{x^{\backslash d}}(z^{\backslash d}) \sum_{x_1^d} u_t^{q,d}(z^d, x^d|x_1^d)q_{1|t}^d(x_1^d|x)$$

$$\stackrel{\triangle}{=} \sum_{d=1}^{\mathcal{D}} \delta_{x^{\backslash d}}(z^{\backslash d})u_t^{q,d}(z^d, x),$$

where $q_{1|t}^d(x_1^d|x) = \sum_{x_1^{\backslash d}} q_{1|t}(x_1|x)$ is the posterior. Therefore, we only need to learn a $\mathcal{D} \times |\mathcal{S}|$-matrix-valued function $U_t^q(x) = (u_t^{q,d}(s, x))_{d,s}$. Then we can update each token in parallel during sampling.

**Training Objective and Sampling Algorithm.** Given the conditional transition rate, the unconditional transition rate can be parameterized by learning the posterior $q_{1|t}^d(x_1^d|x)$. Let $\mathcal{U}([0,1])$ be the uniform distribution on $[0,1]$. A simple learning objective for discrete flow models is given by

$$\mathcal{L}_q = \mathbb{E}_{t \sim \mathcal{U}([0,1]), \mathbf{x}_1 \sim q_1(\mathbf{x}_1), \mathbf{x}_t \sim q_{t|1}(\mathbf{x}_t|\mathbf{x}_1)} \Big[ -\sum_{d=1}^{\mathcal{D}} \log q_{1|t}^{d,\theta}(\mathbf{x}_1^d|\mathbf{x}_t) \Big], \quad (5)$$

which is the cross-entropy between the true posterior $q_{1|t}$ and the estimated posterior $q_{1|t}^\theta$ (Campbell et al., 2024; Gat et al., 2024; Wang et al., 2025).

Additionally, there are some alternative approaches to training the transition rate, including directly maximizing the ELBO in terms of the unconditional transition rate (Shaul et al., 2025) and minimizing a Bregman divergence-type conditional generator matching loss (Holderrieth et al., 2025). It turns out that parameterization through the posterior gives us a unified framework to formulate the discrete guidance not only for discrete flow-based models, but also for other generative models. We include more detailed discussions in Appendix E.1. Moreover, details of the sampling implementation for equation 2 in Algorithm 1 are provided in the Appendix A.

## 3 DISCRETE FLOW GUIDANCE

In this section, we introduce our general guidance framework for modeling discrete data. It can be applied to both discrete flow models and discrete diffusion models (Campbell et al., 2022; Sun et al., 2023; Austin et al., 2021; Lou et al., 2024; Shi et al., 2024; Sahoo et al., 2024). More detailed discussions can be found in Appendix E.

### 3.1 PROBLEM FORMULATION

Consider a CTMC with a probability path $p_t$ associated with a transition rate $u_t^p$. Suppose that the conditional transition rate and the conditional probability path are prespecified (see equation 4 for an example), and we are given a pre-trained posterior $p_{1|t}$ such that the unconditional transition rate $u_t^p(z, x) = \mathbb{E}_{\mathbf{x}_1 \sim p_{1|t}(\mathbf{x}_1|x)}[u_t^p(z, x|\mathbf{x}_1)]$ can generate the probability path $p_t$. We will call the distribution with the pmf $p_1$ the source distribution. Our goal is to generate samples from the target distribution $q_1$ by rectifying the posterior or the unconditional transition rate corresponding to the probability path $p_t$.

### 3.2 EXACT DISCRETE GUIDANCE

To begin with, we first impose the following assumption on the target distribution.

**Assumption 1.** *The target distribution $q_1$ is absolutely continuous with respect to the source distribution $p_1$, i.e., the support of the target probability mass function is a subset of that of the source probability mass function: $\{x \in \mathcal{S}^{\mathcal{D}} \mid q_1(x) > 0\} \subseteq \{x \in \mathcal{S}^{\mathcal{D}} \mid p_1(x) > 0\}$.*

This assumption admits a well-defined density ratio $r(x) = q_1(x)/p_1(x)$ on the support of $p_1$, which is crucial for our formulation of discrete guidance. In the special case setting of conditional generation, the target distribution is the conditional distribution $q_1(x) = p_1(x|y)$ where $y$ denotes the conditioning variable, and the Assumption 1 holds by construction.

Now, assume that the density ratio $r(x) = q_1(x)/p_1(x)$ is known. The following theorem yields the rectified posterior $q_{1|t}$ by reweighting the source posterior $p_{1|t}$ with a guidance term.

**Theorem 1** (Posterior-based guidance)**.** *Suppose that the conditional probability path of the source distribution $p_{t|1}$ and that of the target distribution $q_{t|1}$ are chosen to be the same for any $t \in [0,1]$. Under Assumption 1, the posterior regarding the target distribution has the following form:*

$$q_{1|t}(z^d|x) = \frac{\mathbb{E}_{\mathbf{x}_1^{\backslash d} \sim p(\mathbf{x}_1^{\backslash d} | \mathbf{x}_1^d = z^d, \mathbf{x}_t = x)}[r(\mathbf{x}_1)]}{\mathbb{E}_{\mathbf{x}_1 \sim p_{1|t}(\mathbf{x}_1|x)}[r(\mathbf{x}_1)]} p_{1|t}(z^d|x). \tag{6}$$

*In particular, if $q_1(x) = p_1(x|y)$, we have*

$$q_{1|t}(z^d|x) = p_{1|t}(z^d|x, y) = \frac{p(y|\mathbf{x}_1^d = z^d, \mathbf{x}_t = x)}{p(y|\mathbf{x}_t = x)} p_{1|t}(z^d|x). \tag{7}$$

Here, we only assume that the conditional probability paths are the same. Theorem 1 provides a general guidance framework for both discrete diffusion and flow models, because both of them can be modeled via parameterizing posterior densities. In discrete diffusion models, if we further assume that the transition rates of both forward noising processes are equal, then we can generalize and recover the predictor guidance in Nisonoff et al. (2025), which we state in the following.

**Theorem 2** (Rate-based guidance)**.** *Suppose that the transition rates of the forward noising processes $Q_t^p(x,z)$ and $Q_t^q(x,z)$ are equal for any $t \in [0,1]$. Under Assumption 1, the backward transition rate $u_t^q(z,x)$ of probability $q_t$ has the following form:*

$$u_t^q(z,x) = \frac{\mathbb{E}_{\mathbf{x}_1 \sim p_{1|t}(\mathbf{x}_1|z)}[r(\mathbf{x}_1)]}{\mathbb{E}_{\mathbf{x}_1 \sim p_{1|t}(\mathbf{x}_1|x)}[r(\mathbf{x}_1)]} u_t^p(z,x), \tag{8}$$

*where $u_t^q(z,x)$ and $u_t^p(z,x)$ is the time reversal of $Q_t^p(x,z)$ and $Q_t^q(x,z)$, respectively.*

*Furthermore, if $q_1(x) = p_1(x|y)$, then we have*

$$u_t^p(z,x|y) = \frac{p(y|\mathbf{x}_t = z)}{p(y|\mathbf{x}_t = x)} u_t^p(z,x), \tag{9}$$

*which recovers Equation 2 in Nisonoff et al. (2025).*

Theorem 2 requires the source and target corruption rate matrices to be the same, which is stronger than Theorem 1. Although Theorem 2 gives a transition rate that can generate the target probability path, the transition rate we get is not necessarily the same as what we want ($u_t^q(z,x) = \mathbb{E}_{q_{1|t}(\mathbf{x}_1|x)}[u_t^q(z,x|\mathbf{x}_1)]$). However, Theorem 1 provides a guidance term for the posterior and leads to the desired transition rate $u_t^q(z,x)$. See further discussion in Appendix E.3.

In general cases, the guidance is time-dependent according to Theorem 1 and Theorem 2. However, in the masked diffusion and flow models with $x_1$-independent scheduler, the guidance is time-independent. To see this, note that in those cases the posterior is time-independent (see Theorem 1 of Ou et al. (2025) and Proposition 6 of Gat et al. (2024) for details). Since the guidance is the integral of the product of the density ratio and the posterior, the time-independent posterior can lead to a time-independent guidance.

## 3.3 TRAINING OBJECTIVE

According to Theorem 1, given the density ratio $r(x)$, we can obtain the exact guidance term by learning the conditional expectation of the density ratio $h_t^d(z^d, x) \triangleq$

$\mathbb{E}_{\mathbf{x}_1^{\backslash d} \sim p(\mathbf{x}_1^{\backslash d} | \mathbf{x}_1^d = z^d, \mathbf{x}_t = x)}[r(\mathbf{x}_1)]$. We parameterize the guidance $h_t$ using a neural network. A natural training objective for learning conditional expectation is the Bregman divergence (Banerjee et al., 2005), which is defined as

$$D_F(x \| y) = F(x) - F(y) - \langle \nabla F(y), x - y \rangle, \tag{10}$$

where $F(\cdot)$ is a convex function. The most common choice of $F$ is $F(x) = \|x\|_2^2/2$, and then the Bregman divergence is the $\ell^2$-loss. Unfortunately, as mentioned in Lou et al. (2024), the $\ell^2$-loss does not work well for learning the conditional expectation of a density ratio due to the positive nature of the density ratio. In our work, we take $F(x) = \langle x, \log x \rangle$, and the training objective is given by the following.

$$\mathbb{E}_{t \sim \mathcal{U}([0,1]), \mathbf{x}_1 \sim p_1(\mathbf{x}_1), \mathbf{x}_t \sim p_{t|1}(\mathbf{x}_t | \mathbf{x}_1)} \left[ D_F\left( \mathbf{r}(\mathbf{x}_1) \middle\| \mathbf{h}_t^\theta(\mathbf{x}_1, \mathbf{x}_t) \right) \right]$$

$$= \mathbb{E}_{t \sim \mathcal{U}([0,1]), \mathbf{x}_1 \sim p_1(\mathbf{x}_1), \mathbf{x}_t \sim p_{t|1}(\mathbf{x}_t | \mathbf{x}_1)} \left[ \sum_{d=1}^{\mathcal{D}} h_t^{d,\theta}(\mathbf{x}_1^d, \mathbf{x}_t) - r(\mathbf{x}_1) \log h_t^{d,\theta}(\mathbf{x}_1^d, \mathbf{x}_t) \right] + C \tag{11}$$

$$\triangleq \mathcal{L}_{h,p}(\theta) + C,$$

where $\mathbf{r}(\mathbf{x}_1) = r(\mathbf{x}_1) \mathbb{1}_{\mathcal{D}}$, $\mathbf{h}_t^\theta(\mathbf{x}_1, \mathbf{x}_t) = (h_t^{1,\theta}(\mathbf{x}_1^1, \mathbf{x}_t), \ldots, h_t^{\mathcal{D},\theta}(\mathbf{x}_1^{\mathcal{D}}, \mathbf{x}_t))^\top$ and $C$ is a constant independent of $\theta$.

The loss $\mathcal{L}_{h,p}$ only requires the source distribution data. Additionally, we can utilize samples from the target distribution if they are available to achieve better performance. Similar to Ouyang et al. (2024), we introduce a regularization term allowing us to learn the guidance $h_t$ with sampling $\mathbf{x}_1 \sim q_1$ during training. Notice that the minimizer of the objective in equation 5 is the underlying posterior $q_{1|t}$. Combining with Theorem 1, the exact guidance $h_t$ is the minimizer of the following objective.

$$\mathcal{L}_{h,q}(\theta) = \mathbb{E}_{t \sim \mathcal{U}([0,1]), \mathbf{x}_1 \sim q_1(\mathbf{x}_1), \mathbf{x}_t \sim q_{t|1}(\mathbf{x}_t | \mathbf{x}_1)} \left[ -\sum_{d=1}^{\mathcal{D}} \log \frac{h_t^{d,\theta}(\mathbf{x}_1^d, \mathbf{x}_t)}{\sum_{s \in \mathcal{S}} h_t^{d,\theta}(s, \mathbf{x}_t) p_{1|t}^d(s | \mathbf{x}_t)} \right]. \tag{12}$$

Thus, our final training objective can be written as $\mathcal{L}_h(\theta) = \mathcal{L}_{h,p}(\theta) + \lambda \mathcal{L}_{h,q}(\theta)$, where $\lambda$ denotes the hyperparameter controlling the strength of the regularization.

## 3.4 SAMPLING

After the training stage, we can obtain the learned posterior-based guidance, which is a matrix-valued function $H_t^\theta : \mathcal{S}^{\mathcal{D}} \times [0, 1] \to \mathbb{R}_+^{\mathcal{D} \times |\mathcal{S}|}$ with $[H_t^\theta(x)]_{d,s} = h_t^{d,\theta}(s, x)$. At the sampling stage, given the current state $\mathbf{x}_t$, we first sample $\mathbf{x}_1^d$ from

$$q_{1|t}^{d,\theta}(\cdot | \mathbf{x}_t) = \frac{h_t^{d,\theta}(\cdot, \mathbf{x}_t)}{h_t^{d,\theta}(\mathbf{x}_t)} p_{1|t}^d(\cdot | \mathbf{x}_t)$$

for each $d \in [\mathcal{D}]$, where $h_t^{d,\theta}(x) = \sum_{s \in \mathcal{S}} h_t^{d,\theta}(s, x) p_{1|t}^d(s, x)$. Then, we sample $\mathbf{x}_{t+h}^d$ by always-valid sampling procedure (see equation 14 in Appendix A for details). Specifically, at each step, we input the current state $\mathbf{x}_t$ into both the guidance network and pretrained posterior network, and then multiply the output to get the final transition probability in the target distribution $q_{1|t}^{d,\theta}(\cdot | \mathbf{x}_t)$ to update the current state $\mathbf{x}_t$ like Algorithm 1. The overall sampling algorithm can be found in Algorithm 2 in Appendix A and the detailed design for the general rate-based method proposed in Theorem 2 can be found in Appendix F.

**Remark 1.** *The rate-based guidance involves multiple forward passes, which depend on the number of nonzero elements in the transition rate $u_t^p(z, \mathbf{x}_t)$, $z \neq \mathbf{x}_t$, given the current state $\mathbf{x}_t$. If one use the always-valid sampling algorithm with rate-based guidance Algorithm 3, the guidance entails $\mathcal{D}+1$ function calls in each sampling step, which is still computationally inefficient; see Fig. 2 (d). For predictor guidance, Nisonoff et al. (2025) proposed to use the first-order approximation to estimate the log-ratio $\log \frac{p(y|\mathbf{x}_t=z)}{p(y|\mathbf{x}_t=x)}$ for efficient sampling. To be specific, we give the general form based on our proposed rate-based guidance:*

$$\log \mathbb{E}_{\mathbf{x}_1 \sim p_{1|t}(\mathbf{x}_1 | z)}[r(\mathbf{x}_1)] \approx \log \mathbb{E}_{\mathbf{x}_1 \sim p_{1|t}(\mathbf{x}_1 | x)}[r(\mathbf{x}_1)] + \left\langle z - x, \nabla_x \log \mathbb{E}_{\mathbf{x}_1 \sim p_{1|t}(\mathbf{x}_1 | x)}[r(\mathbf{x}_1)] \right\rangle.$$

*Plugging this into equation 2, the approximated guidance is*

$$u_t^q(z, x) = \exp\left(\left\langle z - x, \nabla_x \log \mathbb{E}_{\mathbf{x}_1 \sim p_{1|t}(\mathbf{x}_1|x)}[r(\mathbf{x}_1)] \right\rangle\right) u_t^p(z, x). \qquad (13)$$

*However, this approximation not only introduces an approximation error but is also unreasonable for discrete data modeling, since the value of the right-hand side depends mainly on the location of $z$ and $x$ in Euclidean space. Fortunately, our proposed posterior-based guidance enables sampling at low computational cost without approximation, requiring only a single forward pass. Table 1 summarizes three types of discrete guidance.*

## 3.5 GUIDANCE FOR PREFERENCE ALIGNMENT

Given the general form of guidance for discrete flow matching, our framework is able to deal with energy-guided sampling (Zhang et al., 2025; Lu et al., 2023) and also preference alignment in reinforcement learning from human feedback (RLHF) (Rector-Brooks et al., 2025). In the following, we provide the formulation for RLHF. Let $\mathbf{c}$ denote the prompt with distribution $p_{\mathbf{c}}$, $\pi_{ref}(\mathbf{o}_1|\mathbf{c})$ denote the reference policy associated with pretrained model $p_{1|t}^d(\mathbf{o}_1^d|\mathbf{c}, \mathbf{o}_t)$. In the RL stage of RLHF (Ouyang et al., 2022), we consider to maximize

$$\mathbb{E}_{\mathbf{c} \sim p_{\mathbf{c}}, \mathbf{o}_1|\mathbf{c} \sim \pi}\Big[\mathcal{R}(\mathbf{c}, \mathbf{o}_1) - \tau \log(\pi(\mathbf{o}_1|\mathbf{c})/\pi_{ref}(\mathbf{o}_1|\mathbf{c}))\Big],$$

where $\tau$ is the temperature that controls the deviation from the reference policy. As mentioned in Rafailov et al. (2023), the above objective has the closed-form maximizer

$$\pi^*(\mathbf{o}_1|\mathbf{c}) = \frac{1}{\mathcal{Z}(\mathbf{c})} \pi_{ref}(\mathbf{o}_1|\mathbf{c}) \exp\left(\frac{\mathcal{R}(\mathbf{c}, \mathbf{o}_1)}{\tau}\right),$$

where $\mathcal{Z}(\mathbf{c}) = \int \pi_{ref}(\mathbf{o}_1|\mathbf{c}) \exp\left(\frac{\mathcal{R}(\mathbf{c}, \mathbf{o}_1)}{\tau}\right) d\mathbf{o}_1$ and $\mathcal{R}(\cdot, \cdot)$ is the reward function which can be rule-based or obtained by training a neural network on the a comparison dataset. Given the reward function, we aim to generate samples $\mathbf{o}_1 \sim \pi^*(\mathbf{o}_1|\mathbf{c})$ conditional on $\mathbf{c}$. If we take $p_1(\cdot) = \pi_{ref}(\cdot|\mathbf{c})$ and $q_1(\cdot) = \pi^*(\cdot|\mathbf{c})$ in Theorem 1, we can obtain a rectified posterior for sampling from $\pi^*$, i.e., we would like to use $\mathbf{h}_t^\theta(\mathbf{o}_1, \mathbf{o}_t, c)$ to approximate $\exp\left(\frac{\mathcal{R}(\mathbf{c}, \mathbf{o}_1)}{\tau}\right)$. We leave the detailed training objective in equation 22 in Appendix G.

## 4 EXPERIMENTS

In this section, we present empirical evidence to verify the efficacy of the proposed framework. In Section 4.1, we conduct proof-of-concept experiments using energy-guided sampling. In Section 4.2, we illustrate the effectiveness of our method on reinforcement learning from human feedback. Built upon a multimodal discrete flow-based model, we demonstrate the effectiveness of our method on text-to-image generation and multimodal understanding benchmark.

## 4.1 SIMULATION RESULTS

**Experimental Setup and Baselines.** We consider a 2-D setting similar to that in Zhang et al. (2025); Lu et al. (2023). The sample space is $\mathcal{S}^{\mathcal{D}} = \{0, 1, \ldots, 32\}^2$. We denote the source distribution as $p_1(x)$ and set the target distribution as its energy-guided version $p_1^{(\gamma)}(x) \propto p_1(x)e^{-\gamma \mathcal{E}(x)}$, where $\gamma \geq 0$ is the guidance strength and $\mathcal{E}(x) = -\log p(y = 1|x)$ is the energy function defined by a given classifier $p(y = 1|x)$. Our goal is to sample from the guided distribution $p_1^{(\gamma)}$ by using either the source transition rate $u_t^p$ or the source posterior $p_{1|t}$. We compare our proposed exact discrete guidance with the predictor guidance introduced in Nisonoff et al. (2025). Note that the density ratio in this case is $r(x_1) = \frac{p_1^{(\gamma)}(x_1)}{p_1(x_1)} = \frac{p^\gamma(y=1|x_1)}{\mathcal{Z}(\gamma)}$, where $\mathcal{Z}(\gamma)$ is a constant that does not depend on $x_1$. Thus, we obtain the posterior-based and rate-based guidance based on Theorem 1 and Theorem 2, respectively. For the predictor guidance proposed by Nisonoff et al. (2025), we use the predictor-guided rate $u_t^{(\gamma)}(z, x) = \left[\frac{\mathbb{E}_{\mathbf{x}_1 \sim p_{1|t}(\mathbf{x}_1|z)} p(y=1|\mathbf{x}_1)}{\mathbb{E}_{\mathbf{x}_1 \sim p_{1|t}(\mathbf{x}_1|x)} p(y=1|\mathbf{x}_1)}\right]^\gamma u_t^p(z, x), \; z \neq x$, which is different from our proposed rate-based guidance; see Appendix E.5 for further discussion.

**Experimental Results.** We sample from the 2-D target distribution $p_1^{(\gamma)}$ using different guidance schemes, as shown in Fig. 2 (a-c). For $\gamma = 0, 3$, both rate-based and posterior-based guidance produce distributions close to the ground truth. When $\gamma = 10, 20$, the distribution generated by the predictor guidance (Nisonoff et al., 2025) differs from the ground-truth distribution. Overall, our posterior-based guidance achieves better performance than its rate-based counterpart, and also yields faster sampling, as illustrated in Fig. 2 (d). We include the experimental details in Appendix H.1 and more experimental results can be found in Appendix H.3.

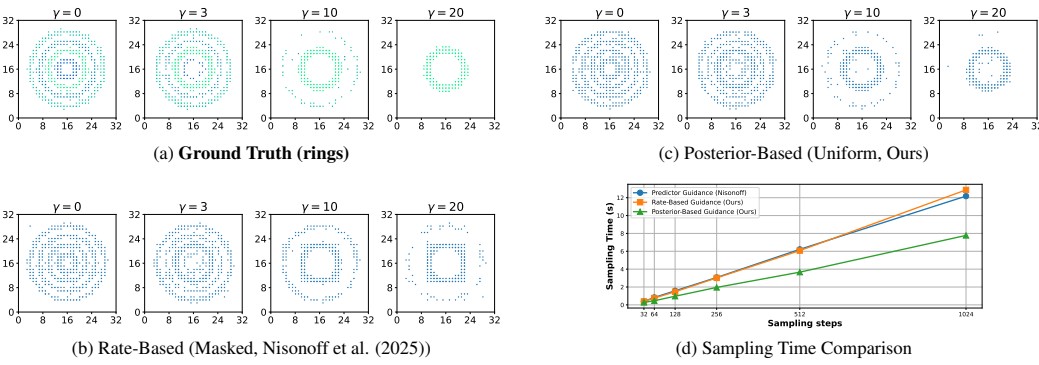

(a) **Ground Truth (rings)**  (c) Posterior-Based (Uniform, Ours)

(b) Rate-Based (Masked, Nisonoff et al. (2025))  (d) Sampling Time Comparison

Figure 2: Comparison of selected sampling results (64 steps) in 2-D experiments and sampling efficiency. (a) Ground truth distribution; (b) rate-based guidance with masked initial distribution (Nisonoff et al., 2025); (c) posterior-based guidance with uniform initial distribution (ours); (d) comparison of sampling times showing posterior-based guidance achieves 1.6x higher sampling speed than rate-based guidance.

## 4.2 RLHF ON MULTIMODAL BENCHMARK

**Experimental Setup and Baselines.** We develop the guidance based on a state-of-the-art discrete flow matching model on multimodal tasks, FUDOKI (Wang et al., 2025). The detailed probability path for FUDOKI can be found in equation 21 and the experimental details can be found in Appendix H.2. For text-to-image generation tasks, we adopt the same training prompts and reward function as Liu et al. (2025b), which utilizes a weighted sum of Pickscore (Kirstain et al., 2023) and GenEval reward (Ghosh et al., 2023) as the reward function. We evaluate the performance on the widely used GenEval Benchmark (Ghosh et al., 2023). For multimodal understanding tasks, we adopt the question and ground-truth answers from SEED (Li et al., 2023a) and the widely used LLM-as-a-judge for assigning reward. We adopt VLMEvalKit (Duan et al., 2024) to evaluate on several multimodal understanding datasets, including POPE (Li et al., 2023b), MME-P (Fu et al., 2023), MMB (Liu et al., 2023c), GQA (Hudson & Manning, 2019), MMMU (Yue et al., 2023), and MM-Vet (Yu et al., 2023).

**Experimental Results.** Results on the GenEval benchmark for text-to-image generation are presented in Table 2. Compared to the setting without guidance, our method achieves improvements on four out of six sub-tasks, highlighting the benefit of guidance in utilizing reward signals. We also provide the ablation studies of the regularization strength in Appendix H.5. The performance on multimodal understanding benchmarks is demonstrated in Table 3. Our guidance framework consistently improves performance. Qualitative results can be found in Fig. 3. Overall, the results confirm the effectiveness of our method for preference alignment across multimodal tasks.

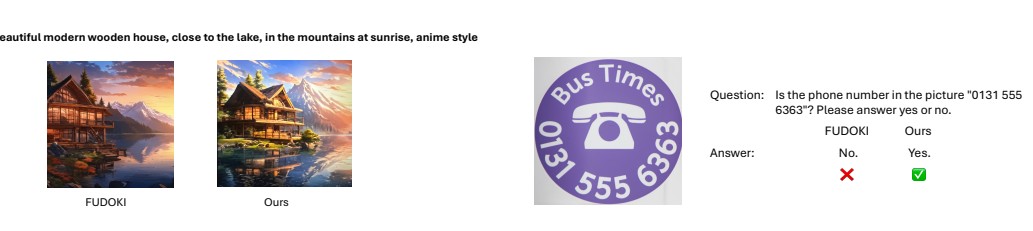

(a) Text-to-Image Generation Results  (b) Multimodal Understanding Results

Figure 3: Qualitative comparison on text-to-image generation and multimodal understanding.

Table 2: Visual Generation Performance on the GenEval Benchmark.

| Type | Method | Single Obj. | Two Obj. | Counting | Colors | Position | Color Attri. | Overall ↑ |
|------|--------|-------------|----------|----------|--------|----------|--------------|-----------|
| Gen. Only | LlamaGen (Sun et al., 2024) | 0.71 | 0.34 | 0.21 | 0.58 | 0.07 | 0.04 | 0.32 |
| | Emu3-Gen (Wang et al., 2024b) | 0.98 | 0.71 | 0.34 | 0.81 | 0.17 | 0.21 | 0.54 |
| | LDM (Rombach et al., 2022) | 0.92 | 0.29 | 0.23 | 0.70 | 0.02 | 0.05 | 0.37 |
| | SDv1.5 (Rombach et al., 2022) | 0.97 | 0.38 | 0.35 | 0.76 | 0.04 | 0.06 | 0.43 |
| | PixArt-alpha (Chen et al., 2024) | 0.98 | 0.50 | 0.44 | 0.80 | 0.08 | 0.07 | 0.48 |
| | SDv2.1 (Rombach et al., 2022) | 0.98 | 0.51 | 0.44 | 0.85 | 0.07 | 0.17 | 0.50 |
| | DALL-E 2 (Ramesh et al., 2022) | 0.94 | 0.66 | 0.49 | 0.77 | 0.10 | 0.19 | 0.52 |
| | SDXL (Podell et al., 2024) | 0.98 | 0.74 | 0.39 | 0.85 | 0.15 | 0.23 | 0.55 |
| | DALL-E 3 (Betker et al.) | 0.96 | 0.87 | 0.47 | 0.83 | 0.43 | 0.45 | 0.67 |
| | SD3-Medium (Esser et al., 2024) | 0.99 | 0.94 | 0.72 | 0.89 | 0.33 | 0.60 | 0.74 |
| Und. + Gen. | SEED-† (Ge et al., 2024) | 0.97 | 0.58 | 0.26 | 0.80 | 0.19 | 0.14 | 0.49 |
| | LWM (Liu et al., 2025a) | 0.93 | 0.41 | 0.46 | 0.79 | 0.09 | 0.15 | 0.47 |
| | ILLUME (Wang et al., 2024a) | 0.99 | 0.86 | 0.45 | 0.71 | 0.39 | 0.28 | 0.61 |
| | TokenFlow-XL (Qu et al., 2024) | 0.95 | 0.60 | 0.41 | 0.81 | 0.16 | 0.24 | 0.55 |
| | Chameleon (Team et al., 2024) | – | – | – | – | – | – | 0.39 |
| | Janus (Wu et al., 2025) | 0.97 | 0.68 | 0.30 | 0.84 | 0.46 | 0.42 | 0.61 |
| | Janus-Pro-1B (Chen et al., 2025) | 0.98 | 0.82 | 0.51 | 0.89 | 0.65 | 0.56 | 0.73 |
| | Show-o (Xie et al., 2025) | 0.95 | 0.52 | 0.49 | 0.82 | 0.11 | 0.28 | 0.53 |
| | Transfusion (Zhou et al., 2025) | – | – | – | – | – | – | 0.63 |
| | UniDisc (Swerdlow et al., 2025) | 0.92 | 0.47 | 0.15 | 0.67 | 0.13 | 0.19 | 0.42 |
| | D-DiT (Li et al., 2024b) | 0.97 | 0.80 | 0.54 | 0.76 | 0.32 | 0.50 | 0.65 |
| | FUDOKI (Wang et al., 2025) | 0.96 | 0.85 | 0.56 | 0.88 | 0.68 | 0.67 | 0.77 |
| | Ours | 0.94 | 0.86 | 0.53 | 0.89 | 0.70 | 0.77 | 0.78 |

*Note:* "Und." = Understanding, "Gen." = Generation. † = models integrating an external pretrained diffusion model. Values exceeding the baseline FUDOKI are highlighted in  gray .

Table 3: Multimodal Understanding Performance on Various Benchmarks.

| Type | Model | # Params | POPE ↑ | MME-P ↑ | MMB ↑ | GQA ↑ | MMMU ↑ | MM-Vet ↑ |
|------|-------|----------|--------|---------|-------|-------|--------|----------|
| Und. Only | LLaVA-Phi-1.5 (Liu et al., 2023b) | 1.3B | 84.1 | 1128.0 | - | 56.5 | 30.7 | - |
| | MobileVLM (Chu et al., 2023) | 1.4B | 84.5 | 1196.2 | 53.2 | 56.1 | - | - |
| | MobileVLM-V2 (Chu et al., 2024) | 1.4B | 84.3 | 1302.8 | 57.7 | 59.3 | - | - |
| | MobileVLM (Chu et al., 2023) | 2.7B | 84.9 | 1288.9 | 59.6 | 59.0 | - | - |
| | MobileVLM-V2 (Chu et al., 2024) | 2.7B | 84.7 | 1440.5 | 63.2 | 61.1 | - | - |
| | LLaVA-Phi (Zhu et al., 2024) | 2.7B | 85.0 | 1335.1 | 59.8 | - | - | 28.9 |
| | LLaVA (Liu et al., 2023b) | 7B | 76.3 | 809.6 | 38.7 | - | - | 25.5 |
| | LLaVA-v1.5 (Liu et al., 2023a) | 7B | 85.9 | 1510.7 | 64.3 | 62.0 | 35.4 | 31.1 |
| | InstructBLIP (Dai et al., 2023) | 7B | - | - | 36.0 | 49.2 | - | 26.2 |
| | Qwen-VL-Chat (Bai et al., 2023) | 7B | - | 1487.5 | 60.6 | 57.5 | - | - |
| | IDEFICS (Laurençon et al., 2023) | 8B | - | - | 48.2 | 38.4 | - | - |
| | Emu3-Chat (Wang et al., 2024b) | 8B | 85.2 | 1244.0 | 58.5 | 60.3 | 31.6 | 37.2 |
| | InstructBLIP (Dai et al., 2023) | 13B | 78.9 | 1212.8 | - | 49.5 | - | 25.6 |
| Und. & Gen. | LaVIT† (Jin et al., 2023) | 7B | - | - | - | 46.8 | - | - |
| | MetaMorph† (Tong et al., 2024) | 8B | - | - | 75.2 | - | - | - |
| | Gemini-Nano-1 (et al., 2024) | 1.8B | - | - | - | - | 26.3 | - |
| | ILLUME (Wang et al., 2024a) | 7B | 88.5 | 1445.3 | 65.1 | - | 38.2 | 37.0 |
| | TokenFlow-XL (Qu et al., 2024) | 13B | 86.8 | 1545.9 | 68.9 | 62.7 | 38.7 | 40.7 |
| | LWM (Liu et al., 2025a) | 7B | 75.2 | - | - | 44.8 | - | 9.6 |
| | VILA-U (Wu et al., 2024) | 7B | 85.8 | 1401.8 | - | 60.8 | - | 33.5 |
| | Chameleon (Team et al., 2024) | 7B | - | - | - | - | 22.4 | 8.3 |
| | Janus (Wu et al., 2025) | 1.5B | 87.0 | 1338.0 | 69.4 | 59.1 | 30.5 | 34.3 |
| | Janus-Pro-1B (Chen et al., 2025) | 1.5B | 86.2 | 1444.0 | 75.5 | 59.3 | 36.3 | 39.8 |
| | Show-o-256 (Xie et al., 2025) | 1.3B | 73.8 | 948.4 | - | 48.7 | 25.1 | - |
| | Show-o-512 (Xie et al., 2025) | 1.3B | 80.0 | 1097.2 | - | 58.0 | 26.7 | - |
| | D-Dit (Li et al., 2024b) | 2.0B | 84.0 | 1124.7 | - | 59.2 | - | - |
| | FUDOKI (Wang et al., 2025) | 1.5B | 86.1 | 1485.4 | 73.9 | 57.6 | 34.3 | 38.0 |
| | Ours | 1.5B | 86.8 | 1492.7 | 74.2 | 58.2 | 35.4 | 38.6 |

*Note:* "Und." = Understanding, "Gen." = Generation. † = models integrating an external pretrained diffusion model. Values exceeding the baseline FUDOKI are highlighted in  gray .

# 5 RELATED WORKS

In this section, we review the main related works to ours. A comprehensive coverage of related literature can be found in Appendix B.

**Guidance.** A line of work has studied guidance for diffusion and flow-matching models in continuous space (Dhariwal & Nichol, 2021; Lu et al., 2023; Ouyang et al., 2024; Feng et al., 2025). Subsequent efforts extend guidance to discrete space (Vignac et al., 2023; Schiff et al., 2025; Li et al., 2024a; Nisonoff et al., 2025), but these methods typically rely on a first-order approximation to reduce the computational cost. However, this approximation is not appropriate in the discrete setting, which our work seeks to overcome.

**Discrete Flow-based Models.** Our work primarily builds on a line of discrete flow-based models (Campbell et al., 2024; Gat et al., 2024; Shaul et al., 2025; Qin et al., 2025). These models construct transition rates that generate the target probability path through marginalizing conditional transition rate, yielding a more comprehensive design space of conditional probability path than discrete diffusion models (Austin et al., 2021; Campbell et al., 2022; Sun et al., 2023; Vignac et al., 2023; Lou et al., 2024) including masked diffusion models (Shi et al., 2024; Sahoo et al., 2024; Ou et al., 2025; Nie et al., 2025c; Zhu et al., 2025; Zhao et al., 2025; Yang et al., 2025). This motivates us to propose posterior-based guidance, which provides a flexible and unified perspective for generative modeling through marginalization.

## 6 CONCLUSION

In this work, we introduce a novel guidance framework for discrete flow-based models. Our approach provides exact guidance with improved sampling efficiency, offering a unified perspective for constructing guidance across generative models. We further propose to train the guidance network by minimizing the Bregman divergence with regularization. The effectiveness of our framework is validated through experiments on energy-guided simulations, preference alignment for text-to-image generation, and multimodal understanding tasks.

## ACKNOWLEDGMENTS

Hongyuan Zha's research is supported in part by Shenzhen Stability Science Program 2023, and National Natural Science Foundation of China (72495131). Fang Fang gratefully acknowledges research support from National Natural Science Foundation of China (72331005). Guang Cheng gratefully acknowledges financial support from the National Science Foundation (NSF–CNS 2247795).

## ETHICS STATEMENT

This work adheres to the ICLR Code of Ethics. Our research does not involve human subjects, sensitive personal data, or experiments that may pose harm to individuals or communities. Publicly available datasets and benchmarks were used in all experiments.

## REPRODUCIBILITY STATEMENT

The code for our implementation can be found in the supplementary materials. For our theoretical result in Theorem 1 and Theorem 2, the assumption is stated in Assumption 1, and the proof can be found in Appendix D.2 and Appendix D.3.

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

## A   SAMPLING ALGORITHMS

Note that the transition probability (Equation 2) is valid only if $h \leq \frac{1}{|u_t^q(x,x)|}$ for each $x \in \mathcal{S}^{\mathcal{D}}$, when we just use its Euler discretization to generate samples. To remove this constraint, following the always-valid sampling procedure introduced in Shaul et al. (2025), given the current state $\mathbf{x}_t$, for each $d \in [\mathcal{D}]$, we first sample $\mathbf{x}_1^d$ from the learned posterior $q_{1|t}^{d,\theta}(\cdot|\mathbf{x}_t)$, and then sample $\mathbf{x}_{t+h}^d$ from

$$
e^{h u_t^{q,d}(\mathbf{x}_t^d, \mathbf{x}_t^d | \mathbf{x}_1^d)} \delta_{\mathbf{x}_t^d}(\cdot) - (1 - e^{h u_t^{q,d}(\mathbf{x}_t^d, \mathbf{x}_t^d | \mathbf{x}_1^d)}) \frac{u_t^{q,d}(\cdot, \mathbf{x}_t^d | \mathbf{x}_1^d)}{u_t^{q,d}(\mathbf{x}_t^d, \mathbf{x}_t^d | \mathbf{x}_1^d)} (1 - \delta_{\mathbf{x}_t^d}(\cdot)), \tag{14}
$$

where we use the first-order approximation

$$
e^{h u_t^{q,d}(\mathbf{x}_t^d, \mathbf{x}_t^d | \mathbf{x}_1^d)} = 1 + h u_t^{q,d}(\mathbf{x}_t^d, \mathbf{x}_t^d | \mathbf{x}_1^d) + o(h).
$$

---

**Algorithm 1** Sampling without Guidance

---

**Require:** pretrained posterior $q_{1|t}$, conditional transition rate $u_t^q(z, x | x_1)$, initial value $x_0$, step size $h$

1: $t \leftarrow 0$
2: $\mathbf{x}_t \leftarrow x_0$
3: **while** $t + h < 1$ **do**
4:     **for** $d = 1, \ldots, \mathcal{D}$ **do**                      ▷ in parallel
5:         Calculate $q_{1|t}^{d,\theta}(x_1^d | \mathbf{x}_t)$
6:         Sample $\mathbf{x}_1^d \sim q_{1|t}^{d,\theta}(\cdot \mid \mathbf{x}_t)$
7:         $\lambda^d \leftarrow \sum_{s \neq \mathbf{x}_t^d} u_t^{q,d}(s, \mathbf{x}_t^d | \mathbf{x}_1^d)$
8:         Sample $Z_{\text{jump}}^d \sim U[0, 1]$
9:         **if** $Z_{\text{jump}}^d \leq 1 - e^{-h \lambda^d}$ **then**
10:            Sample $\mathbf{x}_t^d \sim \frac{u_t^{q,d}(\cdot, \mathbf{x}_t^d | \mathbf{x}_1^d)}{\lambda^d} \left(1 - \delta_{\mathbf{x}_t^d}(\cdot)\right)$
11:         **end if**
12:     **end for**
13:     $t \leftarrow t + h$
14: **end while**
15: $t \leftarrow t - h$
16: **for** $d = 1, \ldots, \mathcal{D}$ **do**                        ▷ in parallel
17:     Sample $\mathbf{x}_1^d \sim q_{1|t}^{d,\theta}(\cdot \mid \mathbf{x}_t)$
18: **end for**
19: **return** $\mathbf{x}_1$

---

## B   ADDITIONAL RELATED WORKS

**Discrete Flow-based Models.** Flow-based models for modeling discrete data were introduced by Campbell et al. (2024); Gat et al. (2024); Shaul et al. (2025); Holderrieth et al. (2025); Qin et al. (2025). Compared to discrete diffusion models (Austin et al., 2021; Campbell et al., 2022; Sun et al., 2023; Lou et al., 2024), the main advantage of discrete flow models is the flexibility of designing the conditional probability path and the conditional transition rate without specifying corruption transition rate; while discrete diffusion models require choosing a forward transition rate with a simple form such that the conditional probability path can be computed easily. Similar to the framework of continuous flow models (Albergo & Vanden-Eijnden, 2023; Liu et al., 2022; Lipman et al., 2023), discrete flow models focus on the conditional path with probability interpolation, which is a convex combination of conditional probabilities. Shaul et al. (2025) proposed the mixture probability path with $x_1$-dependent schedulers and the metric-induced path; the training objective they used is the negative ELBO instead of the cross entropy used in Campbell et al. (2024) and Gat et al. (2024).

---

**Algorithm 2** Sampling with Posterior-Based Guidance

---

**Require:** pretrained posterior $p_{1|t}$, conditional transition rate $u_t^{p,d}(z,x|x_1)$, initial value $x_0$, posterior-based guidance $H_t^\theta$, step size $h$
1: $t \leftarrow 0$
2: $\mathbf{x}_t \leftarrow x_0$
3: **while** $t + h < 1$ **do**
4:     **for** $d = 1, \ldots, \mathcal{D}$ **do**                                                                  ▷ in parallel
5:         Calculate the guided posterior $q_{1|t}^{d,\theta}(x_1^d|\mathbf{x}_t) \propto H_t^\theta(\mathbf{x}_t)_{d,x_1^d} p_{1|t}^d(x_1^d|\mathbf{x}_t)$
6:         Sample $\mathbf{x}_1^d \sim q_{1|t}^{d,\theta}(\cdot \mid \mathbf{x}_t)$
7:         $\lambda^d \leftarrow \sum_{s \neq \mathbf{x}_t^d} u_t^{p,d}(s, \mathbf{x}_t^d|\mathbf{x}_1^d)$
8:         Sample $Z_{\text{jump}}^d \sim U[0,1]$
9:         **if** $Z_{\text{jump}}^d \leq 1 - e^{-h\lambda^d}$ **then**
10:             Sample $\mathbf{x}_t^d \sim \dfrac{u_t^{p,d}(\cdot, \mathbf{x}_t^d|\mathbf{x}_1^d)}{\lambda^d} \left(1 - \delta_{\mathbf{x}_t^d}(\cdot)\right)$
11:         **end if**
12:     **end for**
13:     $t \leftarrow t + h$
14: **end while**
15: $t \leftarrow t - h$
16: **for** $d = 1, \ldots, \mathcal{D}$ **do**                                                                 ▷ in parallel
17:     Calculate the guided posterior $q_{1|t}^{d,\theta}(x_1^d|\mathbf{x}_t) \propto H_t^\theta(\mathbf{x}_t)_{d,x_1^d} p_{1|t}^d(x_1^d|\mathbf{x}_t)$
18:     Sample $\mathbf{x}_1^d \sim q_{1|t}^{d,\theta}(\cdot \mid \mathbf{x}_t)$
19: **end for**
20: **return** $\mathbf{x}_1$

---

Recently, Havasi et al. (2025) introduced a novel discrete flow matching framework with auxiliary process, which supports variable-length sequence generation.

**Discrete Diffusion Models.** Motivated by studies of discrete-time diffusion models in continuous state spaces (Sohl-Dickstein et al., 2015; Ho et al., 2020; Song et al., 2021a; Karras et al., 2022), Austin et al. (2021) and Hoogeboom et al. (2021) proposed a discrete-time framework for training diffusion models in discrete state spaces. Similarly, several authors have developed continuous-time discrete diffusion models through continuous-time Markov chains (Campbell et al., 2022; Benton et al., 2024). Sun et al. (2023) proposed discrete score matching based on the idea of continuous score matching for modeling continuous-time diffusion models (Song et al., 2021b). Meng et al. (2022); Lou et al. (2024) introduced concrete score matching, which aims to identify the time-reversal of the noising process like Sun et al. (2023). Empirically, the masked diffusion models (Shi et al., 2024; Sahoo et al., 2024; Ou et al., 2025) outperform the diffusion models with other noise distributions, such as uniform distribution. Moreover, for masked diffusion models, Ou et al. (2025) showed that the training objective is equivalent to that of any-order autoregressive models (Hoogeboom et al., 2022; Shih et al., 2022), which is also equivalent to the ELBO for discrete flow (see Appendix D.1 in Shaul et al., 2025). The main advantage of discrete diffusion models over (any-order) autoregressive models is parallel sampling. However, supporting variable-length sequence generation is considerably more challenging than in autoregressive models. To overcome this limitation, semi-autoregressive models (Nie et al., 2025c; Arriola et al., 2025) perform block-wise sampling and enable parallel sampling within each block, thus supporting flexible-length sampling.

**Guidance.** Classifier guidance (Dhariwal & Nichol, 2021) and classifier-free guidance (Ho & Salimans, 2021; Zheng et al., 2023) are proposed for conditional generation. Schiff et al. (2025) explored the classifier guidance tailored to discrete diffusion models by decomposing the reversed transition probability for practical purposes. Nie et al. (2025a) applied classifier-free guidance to masked diffusion models for unsupervised pretraining. In this paper, the posterior-based guidance provides a unified perspective for the existing guidance in previous work, including classifier guidance Dhariwal & Nichol (2021), energy-weighted guidance Lu et al. (2023), guidance for transfer learning (Ouyang et al., 2024), and flow matching guidance (Feng et al., 2025) in continuous state space. See

discussion in Appendix E.1. Moreover, thanks to the token-wise transition rate, our discrete guidance only requires training one network, unlike the training-based continuous guidance proposed by (Feng et al., 2025), which requires estimating a normalizing constant by learning a surrogate model.

**Diffusion-based and Flow-based Large Language Models.** Recently, a number of studies have focused on developing large language diffusion models. Several post-training approaches have been proposed to fine-tune discrete diffusion models aligned with human preference in a way similar to autoregressive models via RLHF (Ouyang et al., 2022) or DPO (Rafailov et al., 2023) techniques. LLaDa (Nie et al., 2025c) leverages masked diffusion models and employs negative ELBO for supervised fine-tuning. It supports variable-length sentence generation by semi-autoregressive, and utilizes a deterministic low-confidence remasking technique to enhance performance similar to MaskGIT (Chang et al., 2022). LLaDa 1.5 (Zhu et al., 2025) applied DPO to LLaDa, using an empirical estimator with variance reduction techniques to approximate the log-probability. To extend GRPO to masked diffusion models, d1-LLaDa (Zhao et al., 2025) proposes to estimate log-probability via mean-field approximation and to do one-step unmasking to estimate token-wise log-probability with partially masked prompts. To learn from multi-step denoising information, MMaDa Yang et al. (2025) perturbs sentence outcomes instead of prompts, and conducts policy optimization with a GRPO-type objective, which supports multi-modality. For the large language model based on discrete flow matching, Wang et al. (2025) developed a multi-modal model with a metric-induced probability path and a kinetic-optimal conditional rate (Shaul et al., 2025), achieving highly competitive performance compared to advanced AR-based multi-modal models.

## C BACKGROUND OF CONTINUOUS-TIME MARKOV CHAINS

Continuous-time Markov chains (CTMCs) are a class of continuous-time Markov processes on discrete state spaces. Given a discrete state space $\mathcal{S}^{\mathcal{D}}$, a stochastic process $\mathbf{x}_t$ is CTMC if (a) it has Markov property, that is, given the information of current time, the future and history are independent; (b) given the current time $t$ and state $x \in \mathcal{S}^{\mathcal{D}}$, the transition probability is

$$\mathbb{P}(\mathbf{x}_{t+h} = z | \mathbf{x}_t = x) = \delta_x(z) + u_t(z, x)h + o(h),$$

where $h$ is the step size, and $(u_t(z, x))_{(z,x) \in \mathcal{S}^{\mathcal{D}} \times \mathcal{S}^{\mathcal{D}}}$ is a (time-dynamic) transition rate matrix satisfying

$$u_t(z, x) \geq 0, \text{ for any } z \neq x, \text{ and } \sum_{z \in \mathcal{S}^{\mathcal{D}}} u_t(z, x) = 0 \text{ for any } x \in \mathcal{S}^{\mathcal{D}}.$$

Intuitively, the transition rate $u_t(z, x)$ measures the intensity of transition from the current state $x$ to the target state $z$ at time $t$. This CTMC framework underlies recent advances in generative modeling. Building on the framework of CTMCs, the discrete diffusion models (Austin et al., 2021; Campbell et al., 2022; Sun et al., 2023; Lou et al., 2024) and the discrete flow models (Campbell et al., 2024; Gat et al., 2024; Shaul et al., 2025; Qin et al., 2025) can transport a simple distribution like uniform distribution or just a point mass to a complex data distribution through a sequence of jumps.

## D DERIVATION OF THE GUIDANCE

### D.1 PROOF OF PROPOSITION 3.1 OF (CAMPBELL ET AL., 2024)

We include the proof of Proposition 3.1 of (Campbell et al., 2024) here for completion. We begin by taking the expectation over $p_1$ on both sides of the Kolmogorov equation for $p_{t|1}(x|x_1)$ and $u_t(x_t, z|x_1)$.

$$\partial_t p_t(x) = \mathbb{E}_{\mathbf{x} \sim p_1(\mathbf{x}_1)}\left[\partial_t p_{t|1}(x|\mathbf{x}_1)\right] \quad \text{(definition of } p_t \text{ as marginal over } x_1)$$

$$= \mathbb{E}_{\mathbf{x}_1 \sim p_1(x_1)}\left[\sum_z u_t(x, z|\mathbf{x}_1)\, p_{t|1}(z|\mathbf{x}_1)\right] \quad \text{(Kolmogorov forward equation)}$$

$$= \sum_z \sum_{x_1} p_1(x_1)\, p_{t|1}(z|x_1)\, u_t(x, z|x_1)$$

$$= \sum_z \sum_{x_1} p_t(z)\, p_{1|t}(x_1|z)\, u_t(x, z|x_1) \quad \text{(Bayes' rule)}$$

$$= \sum_z \mathbb{E}_{\mathbf{x} \sim p_{1|t}(\mathbf{x}_1|z)}\left[u_t(x, z|\mathbf{x}_1)\right] p_t(z).$$

We observe that the final expression corresponds to the Kolmogorov equation for a continuous-time Markov chain (CTMC) with marginals $p_t(x)$ and transition rate $\mathbb{E}_{\mathbf{x}_1 \sim p_{1|t}(\mathbf{x}_1|x)}[u_t(z, x|\mathbf{x}_1)]$. This demonstrates that $\mathbb{E}_{\mathbf{x}_1 \sim p_{1|t}(\mathbf{x}_1|x)}[u_t(z, x|\mathbf{x}_1)]$ generates $p_t(x)$.

### D.2 PROOF OF THEOREM 1

*Proof.* By Bayes' formula and the condition $q_{t|1} = p_{t|1}$, we have

$$q_{1|t}(z|x) = \frac{q_{t|1}(x|z)q_1(z)}{q_t(x)} = \frac{p_{t|1}(x|z)q_1(z)}{q_t(x)} = \frac{p_{t|1}(x|z)p_1(z)q_1(z)p_t(x)}{p_t(x)q_t(x)p_1(z)} = \frac{q_1(z)p_t(x)}{q_t(x)p_1(z)}p_{1|t}(z|x)$$

$$= \frac{\frac{q_1(z)}{p_1(z)}}{\sum_{x_1}\frac{q_t(x)}{p_t(x)}q_{1|t}(x_1|x)}p_{1|t}(z|x) = \frac{\frac{q_1(z)}{p_1(z)}}{\sum_{x_1}\frac{q_1(x_1)}{p_1(x_1)}p_{1|t}(x_1|x)}p_{1|t}(z|x)$$

$$= \frac{r(z)}{\mathbb{E}_{\mathbf{x}_1 \sim p_{1|t}(\mathbf{x}_1|x)}[r(\mathbf{x}_1)]}p_{1|t}(z|x),$$

$$(15)$$

which implies that

$$q_{1|t}(z|x) = \frac{r(z)}{\mathbb{E}_{\mathbf{x}_1 \sim p_{1|t}(\mathbf{x}_1|x)}[r(\mathbf{x}_1)]}p(\mathbf{x}_1^{\setminus d} = z^{\setminus d}|\mathbf{x}_t = x, \mathbf{x}_1^d = z^d)p_{1|t}^d(z^d|x).$$

The result equation 6 follows by taking summation over $z^{\setminus d} \in \mathcal{S}^{\mathcal{D}-1}$ for each $d \in [\mathcal{D}]$.

For classifier guidance equation 7, setting $q_1(\cdot) = p_1(\cdot|y)$ in the previous equation, we can obtain that

$$p_{1|t}(z|x, y) = \frac{p_1(z|y)p_t(x)}{p_t(x|y)p_1(z)}p_{1|t}(z|x) = \frac{p(y|\mathbf{x}_1 = z)}{p(y|\mathbf{x}_t = x)}p_{1|t}(z|x)$$

$$= \frac{p(y|\mathbf{x}_1 = z)}{p(y|\mathbf{x}_t = x)}p(\mathbf{x}_1^{\setminus d} = z^{\setminus d}|\mathbf{x}_t = x, \mathbf{x}_1^d = z^d)p_{1|t}^d(z^d|x).$$

Thus, it suffices to show that $p(y|\mathbf{x}_1 = z) = p(y|\mathbf{x}_1 = z, \mathbf{x}_t = x)$. To see this, note that

$$p(y|\mathbf{x}_1 = z) = \frac{p(\mathbf{x}_1 = z, \mathbf{y} = y)}{p_1(z)} = \frac{p(\mathbf{x}_t = x|\mathbf{x}_1 = z)p(\mathbf{x}_1 = z, \mathbf{y} = y)}{p(\mathbf{x}_1 = z, \mathbf{x}_t = x)}$$

$$= \frac{p(\mathbf{x}_t = x|\mathbf{x}_1 = z, \mathbf{y} = y)p(\mathbf{x}_1 = z, \mathbf{y} = y)}{p(\mathbf{x}_1 = z, \mathbf{x}_t = x)} = p(y|\mathbf{x}_1 = z, \mathbf{x}_t = x),$$

which completes the proof. $\qquad\square$

### D.3 PROOF OF THEOREM 2

*Proof.* By Bayes' formula and the condition $q_{t|t+h} = p_{t|t+h}$, we have

$$
\begin{aligned}
q_{t+h|t}(z|x) &= \frac{q_{t|t+h}(x|z)q_{t+h}(z)}{q_t(x)} = \frac{p_{t|t+h}(x|z)q_{t+h}(z)}{q_t(x)} = \frac{p_{t|t+h}(x|z)p_{t+h}(z)q_{t+h}(z)p_t(x)}{p_t(x)q_t(x)p_{t+h}(z)} \\
&= \frac{q_{t+h}(z)p_t(x)}{q_t(x)p_{t+h}(z)}p_{t+h|t}(z|x).
\end{aligned}
\tag{16}
$$

By equation 2, we can obtain that

$$
\begin{aligned}
q_{t+h|t}(z|x) &= \frac{\sum_{x_t}\{\delta_z(x_t) + u_t^q(z,x_t)h + o(h)\}q_t(x_t)p_t(x)}{\sum_{x_t}\{\delta_z(x_t) + u_t^p(z,x_t)h + o(h)\}p_t(x_t)q_t(x)}\Big(\delta_x(z) + u_t^p(z,x)h + o(h)\Big) \\
&= \frac{q_t(z)p_t(x)}{p_t(z)q_t(x)}\Big(\delta_x(z) + u_t^p(z,x)h + o(h)\Big) \\
&= \delta_x(z) + \frac{q_t(z)p_t(x)}{p_t(z)q_t(x)}u_t^p(z,x)h + o(h) \\
&= \delta_x(z) + \frac{\mathbb{E}_{\mathbf{x}_1 \sim p_{1|t}(\mathbf{x}_1|z)}[r(\mathbf{x}_1)]}{\mathbb{E}_{\mathbf{x}_1 \sim p_{1|t}(\mathbf{x}_1|x)}[r(\mathbf{x}_1)]}u_t^p(z,x)h + o(h),
\end{aligned}
$$

where the last equation we use the same trick as the proof of Theorem 1. Then, the transition rate of the probability path $q_t$ is

$$
u^q(z,x) = \lim_{h\to 0^+} \frac{q_{t+h|t}(z|x) - \delta_x(z)}{h} = \frac{\mathbb{E}_{\mathbf{x}_1 \sim p_{1|t}(\mathbf{x}_1|z)}[r(\mathbf{x}_1)]}{\mathbb{E}_{\mathbf{x}_1 \sim p_{1|t}(\mathbf{x}_1|x)}[r(\mathbf{x}_1)]}u_t^p(z,x).
$$

For classifier guidance, considering $q_1(x_1) = p_1(x|y)$, we have

$$
u^q(z,x) = u^p(z,x|y) = \frac{p_t(z|y)p_t(x)}{p_t(z)p_t(x|y)}u_t^p(z,x) = \frac{p(y|\mathbf{x}_t = z)}{p(y|\mathbf{x}_t = x)}u_t^p(z,x),
$$

which completes the proof. □

## E IN-DEPTH DISCUSSIONS OF THE PROPOSED GUIDANCE FRAMEWORK

### E.1 UNIFYING GUIDANCE THROUGH MARGINALIZATION TRICK

In this subsection, we provide the guidance formulation for continuous score matching, continuous flow matching, concrete score matching, and discrete flow matching. We assume that the conditional probability path of the source data distribution is the same as that of the target data distribution.

#### E.1.1 CONTINUOUS SCORE MATCHING

We will use the convention of using $t = 0$ for data distribution and $t = T$ for noise in continuous diffusion models. With a little abuse of notation, denote $r(z) = \frac{q_0(z)}{p_0(z)}$. From equation 15, the score

function for sampling from $q_0$ is

$$\nabla_x \log q_t(x) = \mathbb{E}_{\mathbf{x}_0 \sim q_{0|t}(\mathbf{x}_0|x)}[\nabla_x \log q_{t|0}(x|\mathbf{x}_0)]$$

$$= \int_{x_0} q_{0|t}(x_0|x)\nabla_x \log q_{t|0}(x|x_0)\mathrm{d}x_0$$

$$= \int_{x_0} \frac{r(x_0)}{\mathbb{E}_{\mathbf{x}_0 \sim p_{0|t}(\mathbf{x}_0|x)}[r(\mathbf{x}_0)]} p_{0|t}(x_0|x)\nabla_x \log p_{t|0}(x|x_0)\mathrm{d}x_0$$

$$= \nabla_x \log p_t(x) + \int_{x_0} \frac{r(x_0) - \mathbb{E}_{\mathbf{x}_0 \sim p_{0|t}(\mathbf{x}_0|x)}[r(\mathbf{x}_0)]}{\mathbb{E}_{\mathbf{x}_0 \sim p_{0|t}(\mathbf{x}_0|x)}[r(\mathbf{x}_0)]} p_{0|t}(x_0|x)\nabla_x \log p_{t|0}(x|x_0)\mathrm{d}x_0$$

$$= \nabla_x \log p_t(x) + \int_{x_0} \frac{\frac{q_0(x_0)}{p_0(x_0)} - \frac{q_t(x)}{p_t(x)}}{\mathbb{E}_{\mathbf{x}_0 \sim p_{0|t}(\mathbf{x}_0|x)}[r(\mathbf{x}_0)]} \cdot \frac{p_0(x_0)\nabla_x p_{t|0}(x|x_0)}{p_t(x)}\mathrm{d}x_0$$

$$= \nabla_x \log p_t(x) + \frac{\frac{\nabla_x p_q(x)}{p_t(x)} - \frac{q_t(x)\nabla_x p_t(x)}{p_t(x)^2}}{\mathbb{E}_{\mathbf{x}_0 \sim p_{0|t}(\mathbf{x}_0|x)}[r(\mathbf{x}_0)]}$$

$$= \nabla_x \log p_t(x) + \nabla_x \log \mathbb{E}_{\mathbf{x}_0 \sim p_{0|t}(\mathbf{x}_0|x)}[r(\mathbf{x}_0)],$$

where we use the facts $q_{t|0} = p_{t|0}$ and $\mathbb{E}_{\mathbf{x}_0 \sim p_{0|t}(\mathbf{x}_0|x)}[r(\mathbf{x}_0)] = \frac{q_t(x)}{p_t(x)}$. This result recovers the guidance given by Ouyang et al. (2024). By setting $q_t(x) = p_t(x|y)$, we also recover the classifier guidance $\nabla_x \log q_t(x) = \nabla_x \log p_t(x) + \nabla_x \log p(y|\mathbf{x}_t = x)$.

### E.1.2 CONTINUOUS FLOW MATCHING

Denote $v_t(x)$ and $v_t(x \mid x_1)$ as the marginal and conditional velocity fields, respectively. Assuming the conditional kernels satisfy $q_{t|1}(x \mid x_1) = p_{t|1}(x \mid x_1)$, and defining $r(x_1) := q_1(x_1)/p_1(x_1)$, we have

$$v_t^q(x) = \mathbb{E}_{\mathbf{x}_1 \sim q_{1|t}(\mathbf{x}_1|x)}[v_t(x \mid \mathbf{x}_1)]$$

$$= \mathbb{E}_{\mathbf{x}_1 \sim p_{1|t}(\mathbf{x}_1|x)}[v_t(x \mid \mathbf{x}_1)\frac{q_{1|t}(\mathbf{x}_1 \mid x)}{p_{1|t}(\mathbf{x}_1 \mid x)}]$$

$$= \mathbb{E}_{\mathbf{x}_1 \sim p_{1|t}(\mathbf{x}_1|x)}[v_t(x \mid \mathbf{x}_1)\frac{r(\mathbf{x}_1)}{\mathbb{E}_{\mathbf{x}_1 \sim p_{1|t}(\mathbf{x}_1|x)}[r(\mathbf{x}_1)]}]$$

$$= v_t^p(x) + \int_{x_1} \left(\frac{r(x_1)}{\mathbb{E}_{\mathbf{x}_1 \sim p_{1|t}(\mathbf{x}_1|x)}[r(\mathbf{x}_1)]} - 1\right)v_t(x \mid x_1)\,p_{1|t}(x_1 \mid x)\,\mathrm{d}x_1.$$

which coincides the exact guidance (Feng et al., 2025) for continuous flow matching.

### E.1.3 DISCRETE DIFFUSION MODELS

Denote the concrete score (Meng et al., 2022; Lou et al., 2024) $s_t(x) = \left(\frac{p_t(z)}{p_t(x)}\right)_{z^d \neq x^d, z^{\backslash d} = x^{\backslash d}}$, which is a $\mathcal{D}(|\mathcal{S}| - 1)$-dimensional vector. Then, for any $z, x \in \mathcal{S}^{\mathcal{D}}$ satisfying $z^d \neq x^d, z^{\backslash d} = x^{\backslash d}$, we have

$$s_t^q(x)_z = \mathbb{E}_{\mathbf{x}_1 \sim q_{1|t}(\mathbf{x}_1|x)}\left[\frac{q_{t|1}(z|\mathbf{x}_1)}{q_{t|1}(x|\mathbf{x}_1)}\right]$$

$$= \mathbb{E}_{\mathbf{x}_1 \sim p_{1|t}(\mathbf{x}_1|x)}\left[\frac{p_{t|1}(z|\mathbf{x}_1)q_{1|t}(\mathbf{x}_1|x)}{p_{t|1}(x|\mathbf{x}_1)p_{1|t}(\mathbf{x}_1|x)}\right]$$

$$= s_t^p(x)_z + \sum_{x_1} \left(\frac{r(x_1)}{\mathbb{E}_{\mathbf{x}_1 \sim p_{1|t}(\mathbf{x}_1|x)}[r(\mathbf{x}_1)]} - 1\right)\prod_{d=1}^{\mathcal{D}} \frac{p_{t|1}^d(z^d|x_1^d)}{p_{t|1}^d(x^d|x_1^d)}p_{1|t}(x_1|x)$$

$$= s_t^p(x)_z + \sum_{x_1^d} \left(\frac{\mathbb{E}_{\mathbf{x}_1^{\backslash d} \sim p(\mathbf{x}_1^{\backslash d}|\mathbf{x}_1^d = x_1^d, \mathbf{x}_t = x)}[r(\mathbf{x}_1)]}{\mathbb{E}_{\mathbf{x}_1 \sim p_{1|t}(\mathbf{x}_1|x)}[r(\mathbf{x}_1)]} - 1\right)\frac{p_{t|1}^d(z^d|x_1^d)}{p_{t|1}^d(x^d|x_1^d)}p_{1|t}^d(x_1^d|x),$$

where we use $t = 1$ for the data distribution. Following the discussion in Appendix E.3, the formulation above can be viewed as a special case of discrete flow matching.

### E.1.4 DISCRETE FLOW MATCHING

Similar to the previous formulation, we can obtain that

$$u_t^{q,d}(z^d,x) = u_t^{p,d}(z^d,x) + \sum_{x_1^d}\Big(\frac{\mathbb{E}_{\mathbf{x}_1^{\backslash d}\sim p(\mathbf{x}_1^{\backslash d}|\mathbf{x}_1^d=x_1^d,\mathbf{x}_t=x)}[r(\mathbf{x}_1)]}{\mathbb{E}_{\mathbf{x}_1\sim p_{1|t}(\mathbf{x}_1|x)}[r(\mathbf{x}_1)]}-1\Big)u_t^{p,d}(z^d,x^d|x_1^d)p_{1|t}^d(x_1^d|x).$$

To give a specific example, we consider the $x_1$-independent mixture path and its kinetic-optimal transition rate (see Appendix C of Shaul et al., 2025):

$$p_{t|1}^d(x^d|x_1^d) = q_{t|1}^d(x^d|x_1^d) = \kappa_t \delta_{x_1^d}(x^d) + (1-\kappa_t)p_0(x^d);$$

$$u_t^{q,d}(z^d,x^d|x_1^d) = u_t^{p,d}(z^d,x^d|x_1^d) = \frac{\dot{\kappa}_t}{1-\kappa_t}(\delta_{x_1^d}(z^d) - \delta_{x^d}(z^d)).$$

Then, we have

$$u_t^{q,d}(z^d,x) = u_t^{p,d}(z^d,x) + \frac{\dot{\kappa}_t}{1-\kappa_t}\Big(\frac{\mathbb{E}_{\mathbf{x}_1^{\backslash d}\sim p(\mathbf{x}_1^{\backslash d}|\mathbf{x}_1^d=z^d,\mathbf{x}_t=x)}[r(\mathbf{x}_1)]}{\mathbb{E}_{\mathbf{x}_1\sim p_{1|t}(\mathbf{x}_1|x)}[r(\mathbf{x}_1)]}-1\Big)p_{1|t}^d(z^d|x)$$

$$= \frac{\mathbb{E}_{\mathbf{x}_1^{\backslash d}\sim p(\mathbf{x}_1^{\backslash d}|\mathbf{x}_1^d=z^d,\mathbf{x}_t=x)}[r(\mathbf{x}_1)]}{\mathbb{E}_{\mathbf{x}_1\sim p_{1|t}(\mathbf{x}_1|x)}[r(\mathbf{x}_1)]}u_t^{p,d}(z^d,x)$$

$$+ \frac{\dot{\kappa}_t}{1-\kappa_t}\Big(\frac{\mathbb{E}_{\mathbf{x}_1^{\backslash d}\sim p(\mathbf{x}_1^{\backslash d}|\mathbf{x}_1^d=z^d,\mathbf{x}_t=x)}[r(\mathbf{x}_1)]}{\mathbb{E}_{\mathbf{x}_1\sim p_{1|t}(\mathbf{x}_1|x)}[r(\mathbf{x}_1)]}-1\Big)\delta_{x^d}(z^d),$$

which implies that $u_t^{q,d}(z^d,x)$ is an affine combination of $u_t^{p,d}(z^d,x)$ and $-\frac{\dot{\kappa}_t}{1-\kappa_t}\delta_{x^d}(z^d)$ in this case.

### E.2 DISCRETE GUIDANCE FOR DISCRETE-TIME FRAMEWORK

In this subsection, we discuss the discrete guidance in the settings of the discrete-time Markov chain (e.g., D3PM, Austin et al., 2021), which is developed by Vignac et al. (2023); Schiff et al. (2025); Li et al. (2024a). For convenience, we also use $t = 1$ for data distribution and $t = 0$ for initial distribution. Assume that $p_{s|t} = q_{s|t}$ for each $s < t$. By Equation 16, the transition probability of $q$ in the sampling process is

$$q_{t+h|t}(z|x) = \frac{q_{t+h}(z)/p_{t+h}(z)}{q_t(x)/p_t(x)}p_{t+h|t}(z|x) = \frac{\mathbb{E}_{\mathbf{x}_1\sim p_{1|t+h}(\mathbf{x}_1|z)}[r(\mathbf{x}_1)]}{\mathbb{E}_{\mathbf{x}_1\sim p_{1|t}(\mathbf{x}_1|x)}[r(\mathbf{x}_1)]}p_{t+h|t}(z|x),$$

which is similar to the soft optimal policy introduced in Li et al. (2024a). By taking $q_{t+h|t}(z|x) = p_{t+h|t}(z|x,y)$, we recover the classifier guidance in Vignac et al. (2023) and Schiff et al. (2025) with unit guidance strength.

Following the proof of Theorem 2, the rate-based guidance can be viewed as a limiting form of the discrete guidance in discrete time settings. In addition, the discrete-time guidance has a similar challenge to the rate-based guidance; that is, it requires multiple function evaluations in each sampling step. For solving this problem, Vignac et al. (2023); Schiff et al. (2025) employ the first-order approximation in practice. As pointed out in Remark 1, such approximation-based approaches are inappropriate for discrete data intuitively.

### E.3 COMPARISON BETWEEN POSTERIOR-BASED GUIDANCE AND RATE-BASED GUIDANCE

Theorem 1 and Theorem 2 present the discrete guidance based on the posterior and the transition rate, respectively. The later one generalizes the discrete guidance of Nisonoff et al. (2025), which requires a stronger condition than the posterior-based guidance in Theorem 1. To better understand the conditions in both theorems, we will analyze the relationship between discrete diffusion and discrete flow models. In the following, we consider two different states $x \neq z$.

**Both Guidance Yield the Same Transition Rates in Discrete Diffusion Models.** Consider a discrete diffusion model with the corruption transition rate $Q_t^q(x,z)$. Here, we assume that $q_1$ is the

density of the data distribution. Following Campbell et al. (2022), the time reversal of $Q_t^q(x, z)$ is

$$u_t^q(z, x) = \sum_{x_1} Q_t^q(x, z) \frac{q_{t|1}(z|x_1)}{q_{t|1}(x|x_1)} q_{1|t}(x_1|x) = \mathbb{E}_{\mathbf{x}_1 \sim q_{1|t}(\mathbf{x}_1|\mathbf{x}_t=x)} \left[ Q_t^q(x, z) \frac{q_{t|1}(z|\mathbf{x}_1)}{q_{t|1}(x|\mathbf{x}_1)} \right]. \quad (17)$$

Since $u_t^q(z, x|x_1) \triangleq Q_t^q(x, z) \frac{q_{t|1}(z|x_1)}{q_{t|1}(x|x_1)}$ is a conditional transition rate that can generate the conditional path $q_{t|1}$ (see Appendix H.1 of Campbell et al., 2024), the discrete diffusion models can be viewed as a special case of discrete flow models. Therefore, the condition $Q_t^q(x, z) = Q_t^p(x, z)$ as in Theorem 2 actually implies that we take the same specific conditional transition rate $u_t^p(z, x|x_1) = u_t^q(z, x|x_1) = Q_t^q(x, z) \frac{q_{t|1}(z|x_1)}{q_{t|1}(x|x_1)}$ in discrete flow models.

**They Might Offer Different Marginal Transition Rates for Sampling from Target Distribution.**
Suppose that we are given conditional probability path $q_{t|1} = p_{t|1}$ and conditional transition rate $u_t^q(z, x|x_1) = u_t^p(z, x|x_1)$. What we want to know is whether there exists a common corruption transition rate $Q_t(x, z)$ such that $u_t^q(z, x)$ and $u_t^p(z, x)$ are the reverse transition rates of $Q_t(x, z)$ under the data distributions $q_1$ and $p_1$, respectively. If exists, by the time reversal formula (see, e.g. Sun et al., 2023; Lou et al., 2024), we have

$$Q_t(x, z) = \frac{p_t(x)}{p_t(z)} u_t^p(z, x) = \sum_{x_1} \frac{p_t(x)}{p_t(z)} u_t^p(z, x|x_1) p_{1|t}(x_1|x)$$

$$= \sum_{x_1} \frac{p_t(x)}{p_t(z)} Q_t^p(x, z|x_1) \frac{p_{t|1}(z|x_1)}{p_{t|1}(x|x_1)} p_{1|t}(x_1|x)$$

$$= \frac{q_t(x)}{q_t(z)} u_t^q(z, x) = \sum_{x_1} \frac{q_t(x)}{q_t(z)} u_t^q(z, x|x_1) q_{1|t}(x_1|x)$$

$$= \sum_{x_1} \frac{q_t(x)}{q_t(z)} Q_t^q(x, z|x_1) \frac{q_{t|1}(z|x_1)}{q_{t|1}(x|x_1)} q_{1|t}(x_1|x),$$

which implies that

$$\sum_{x_1} Q_t^p(x, z|x_1) p_{1|t}(x_1|z) = \sum_{x_1} Q_t^q(x, z|x_1) q_{1|t}(x_1|z),$$

where $Q_t^p(x, z|x_1) = Q_t^q(x, z|x_1) = u_t^q(z, x|x_1) \frac{q_{t|1}(x|x_1)}{q_{t|1}(z|x_1)}$ is the time reversal of $u_t^q(z, x|x_1) = u_t^p(z, x|x_1)$. However, since the posteriors $p_{1|t}$ and $q_{1|t}$ are different, the above equation generally does not hold, unless $Q_t^p(x, z|x_1) = Q_t^q(x, z|x_1)$ is independent of $x_1$. If we use the rate-based guidance of Theorem 2 with the marginal transition rate $u_t^p(z, x) = \mathbb{E}_{\mathbf{x}_1 \sim p_{1|t}(\mathbf{x}_1|x)}[u_t^p(z, x|\mathbf{x}_1)]$ that is used to sample from source data distribution, then we implicitly set the corruption rate $Q_t^q(x, z) = Q_t^p(x, z) = \mathbb{E}_{\mathbf{x}_1 \sim p_{1|t}(\mathbf{x}_1|x)}[u_t^p(z, x|\mathbf{x}_1)] \frac{p_t(x)}{p_t(z)}$ in discrete diffusion models. In this scenario, the reversed transition rate for generating probability path $q_t$ is

$$\mathbb{E}_{\mathbf{x}_1 \sim p_{1|t}(\mathbf{x}_1|x)}[u_t^p(z, x|\mathbf{x}_1)] \frac{p_t(x)q_t(z)}{p_t(z)q_t(x)},$$

which is not equal to the desired transition rate $\mathbb{E}_{\mathbf{x}_1 \sim q_{1|t}(\mathbf{x}_1|x)}[u_t^q(z, x|\mathbf{x}_1)]$ which we obtained based on Theorem 1 in general. The main reason is that $\mathbb{E}_{\mathbf{x}_1 \sim q_{1|t}(\mathbf{x}_1|x)}[u_t^q(z, x|\mathbf{x}_1)]$ and $\mathbb{E}_{\mathbf{x}_1 \sim p_{1|t}(\mathbf{x}_1|x)}[u_t^p(z, x|\mathbf{x}_1)]$ do not share the same time reversal as mentioned above. Moreover, if we choose a conditional path $u_t^q(z, x|x_1)$ such that $u_t^q(z, x|x_1) \frac{q_{t|1}(x|x_1)}{q_{t|1}(z|x_1)}$ is independent of $x_1$, then these two methods are equivalent. Formally, we have the following proposition.

**Proposition 1.** *Assume that $u_t^q(z, x|x_1) \frac{q_{t|1}(x|x_1)}{q_{t|1}(z|x_1)}$ is unrelated to $x_1$ for two states $x \neq z$. Let $Q_t(x, z) = u_t^q(z, x|x_1) \frac{q_{t|1}(x|x_1)}{q_{t|1}(z|x_1)}$ be the transition rate of noising process. Under the same conditions of Theorem 1, we have $Q_t(x, z) \frac{p_t(z)}{p_t(x)} = \mathbb{E}_{\mathbf{x}_1 \sim p_{t|1}(\mathbf{x}_1|x)}[u_t^p(z, x|\mathbf{x}_1)]$ and $Q_t(x, z) \frac{q_t(z)}{q_t(x)} = \mathbb{E}_{\mathbf{x}_1 \sim q_{t|1}(\mathbf{x}_1|x)}[u_t^q(z, x|\mathbf{x}_1)]$, which means that two methods are equivalent.*

*Proof.* We only consider the case of $p_t$. Note that

$$
\begin{aligned}
\mathbb{E}_{\mathbf{x}_1 \sim p_{t|1}(\mathbf{x}_1|x)}[u_t^p(z, x|\mathbf{x}_1)] &= \sum_{x_1} u_t^p(z, x|x_1) p_{1|t}(x_1|x) \\
&= \sum_{x_1} Q_t^p(x, z) \frac{p_{t|1}(z|x_1)}{p_{t|1}(x|x_1)} p_{1|t}(x_1|x) \\
&= Q_t^p(x, z) \frac{p_t(z)}{p_t(x)},
\end{aligned}
$$

which completes the proof. $\qquad\square$

In summary, the condition $Q_t^q(x, z) = Q_t^p(x, z)$ in Theorem 2 is more restrictive than Theorem 1 if our goal is to use the desired transition rate $u_t^q(z, x) = \mathbb{E}_{\mathbf{x}_1 \sim q_{1|t}(\mathbf{x}_1|x)}[u_t^q(z, x|\mathbf{x}_1)]$ in the sampling stage.

**From Posterior-based Guidance to Rate-based Guidance.** For completeness, we also give a simple proof of Theorem 2 based on the result of Theorem 1. Notice that, if $Q_t^q(x, z) = Q_t^p(x, z)$ (which implies that $p_{t|1} = q_{t|1}$), by equation 17, we have

$$
\begin{aligned}
\frac{u_t^q(z, x)}{u_t^p(z, x)} &= \frac{\mathbb{E}_{\mathbf{x}_1 \sim q_{1|t}(\mathbf{x}_1|x)}\left[\frac{q_{t|1}(z|\mathbf{x}_1)}{q_{t|1}(x|\mathbf{x}_1)}\right]}{\mathbb{E}_{\mathbf{x}_1 \sim p_{1|t}(\mathbf{x}_1|x)}\left[\frac{p_{t|1}(z|\mathbf{x}_1)}{p_{t|1}(x|\mathbf{x}_1)}\right]} = \frac{\mathbb{E}_{\mathbf{x}_1 \sim p_{1|t}(\mathbf{x}_1|x)}\left[\frac{q_{1|t}(\mathbf{x}_1|x) q_{t|1}(z|\mathbf{x}_1)}{p_{1|t}(\mathbf{x}_1|x) q_{t|1}(x|\mathbf{x}_1)}\right]}{\mathbb{E}_{\mathbf{x}_1 \sim p_{1|t}(\mathbf{x}_1|x)}\left[\frac{p_{t|1}(z|\mathbf{x}_1)}{p_{t|1}(x|\mathbf{x}_1)}\right]} \\
&= \frac{\mathbb{E}_{\mathbf{x}_1 \sim p_{1|t}(\mathbf{x}_1|x)}\left[\frac{q_t(z) q_{1|t}(\mathbf{x}_1|z)}{q_t(x) p_{1|t}(\mathbf{x}_1|x)}\right]}{\mathbb{E}_{\mathbf{x}_1 \sim p_{1|t}(\mathbf{x}_1|x)}\left[\frac{p_{t|1}(z|\mathbf{x}_1)}{p_{t|1}(x|\mathbf{x}_1)}\right]} = \frac{q_t(z)/q_t(x)}{p_t(z)/p_t(x)} = \frac{q_t(z)/p_t(x)}{q_t(z)/p_t(x)} \\
&= \frac{\mathbb{E}_{\mathbf{x}_1 \sim p(\mathbf{x}_1|z)}[r(\mathbf{x}_1)]}{\mathbb{E}_{\mathbf{x}_1 \sim p(\mathbf{x}_1|x)}[r(\mathbf{x}_1)]},
\end{aligned}
$$

where the third equation we use $\frac{q_{1|t}(\mathbf{x}_1|x)}{p_{1|t}(\mathbf{x}_1|x)} = \frac{q_1(\mathbf{x}_1) p_t(x)}{p_1(\mathbf{x}_1) q_t(x)}$ in equation 15 and the fourth equation we use the marginalization trick in the concrete score matching (Meng et al., 2022; Lou et al., 2024). This completes the proof.

### E.4 CLASSIFICATION OF DISCRETE GUIDANCE

As discussed in Appendix E.2, the rate-based guidance can be interpreted as the limiting case of the discrete guidance in the discrete-time framework, since all of them require the condition: $p_{s|t}(x_s|x_t) = q_{s|t}(x_s|x_t)$ for any $0 \le s < t \le 1$ (here, we use $t = 1$ for data distribution). In contrast, the posterior-based guidance (Theorem 1) only imposes the condition of conditional probability paths, offering greater flexibility and interpretability for discrete flow models. Accordingly, discrete guidance can be divided into two categories, as summarized in Table 4.

Table 4: Classification of discrete guidance. (Data distribution: $t = 1$)

| Assumption | Time Framework | Discrete Guidance | Target Transition Rate for Discrete Flow Models |
|---|---|---|---|
| $p_{t|1}(x_t|x_1) = q_{t|1}(x_t|x_1)$ for any $0 \le t \le 1$ | Continuous | Posterior-Based Guidance (Theorem 1) | $\mathbb{E}_{\mathbf{x}_1 \sim q_{1|t}(\mathbf{x}_1|x)}[u_t(z, x|\mathbf{x}_1)]$ |
| $p_{s|t}(x_s|x_t) = q_{s|t}(x_s|x_t)$ for any $0 \le s < t \le 1$ | Continuous | Rate-Based Guidance (Theorem 2 & Nisonoff et al. (2025)) | $\mathbb{E}_{\mathbf{x}_1 \sim p_{1|t}(\mathbf{x}_1|x)}[u_t(z, x|\mathbf{x}_1)]\frac{p_t(x)q_t(z)}{p_t(z)q_t(x)}$ |
| | Discrete | Vignac et al. (2023)& Schiff et al. (2025)& Li et al. (2024a) | — |

### E.5 DISCUSSION ON THE GUIDANCE STRENGTH

In this subsection, we focus on the settings considered in Zheng et al. (2023); Zhang et al. (2025). Our goal is to generate data from the guided distribution $p_1^{(\gamma)}(x|y) \propto p_1(x) p^\gamma(y|x)$, where $\gamma \ge 0$ is the guidance strength. Lemma 4.10 in Zhang et al. (2025) provides the comparison between contrastive energy prediction Lu et al. (2023) and classifier guidance Dhariwal & Nichol (2021); Ho & Salimans (2021). Here, we compare the rate-based guidance with the predictor guidance Nisonoff et al. (2025) under different guidance strengths.

The guided transition rate with predictor guidance strength $\gamma$ is

$$u_t^{(\gamma)}(z, x|y) = \left[\frac{p_t(y|z)}{p_t(y|x)}\right]^\gamma u_t(z, x) \tag{18}$$

for $z \neq x$. Here, $u_t$ is a transition rate that can generate $p_t(x)$. In general, if $\gamma \neq 1$, the above guided rate $u_t^{(\gamma)}$ cannot generate $p_t^{(\gamma)}(x|y) = \frac{p_t(x)p_t^\gamma(y|x)}{\mathcal{Z}_t(y;\gamma)}$, where $\mathcal{Z}_t(y;\gamma) = \sum_x p_t(x)p_t^\gamma(y|x)$. To see this, we try to verify the Kolmogorov forward equation:

$$\sum_{x\neq z} u_t^{(\gamma)}(z, x|y)\frac{p_t(x)p_t^\gamma(y|x)}{\mathcal{Z}_t(y;\gamma)} - \sum_{x\neq z} u_t^{(\gamma)}(x, z|y)\frac{p_t(z)p_t^\gamma(y|z)}{\mathcal{Z}_t(y;\gamma)}$$

$$= \sum_{x\neq z}\left[\frac{p_t(y|z)}{p_t(y|x)}\right]^\gamma u_t(z, x)\frac{p_t(x)p_t^\gamma(y|x)}{\mathcal{Z}_t(y;\gamma)} - \sum_{x\neq z}\left[\frac{p_t(y|x)}{p_t(y|z)}\right]^\gamma u_t(x, z)\frac{p_t(z)p_t^\gamma(y|z)}{\mathcal{Z}_t(y;\gamma)}$$

$$= \sum_{x\neq z} u_t(z, x)\frac{p_t(x)p_t^\gamma(y|z)}{\mathcal{Z}_t(y;\gamma)} - \sum_{x\neq z} u_t(x, z)\frac{p_t(z)p_t^\gamma(y|x)}{\mathcal{Z}_t(y;\gamma)}$$

$$= \frac{p_t^\gamma(y|z)}{\mathcal{Z}_t(y;\gamma)}\partial_t p_t(z) + p_t(z)\sum_{x\neq z} u_t(x, z)\frac{[p_t^\gamma(y|z) - p_t^\gamma(y|x)]}{\mathcal{Z}_t(y;\gamma)}$$

$$\neq \frac{p_t^\gamma(y|z)}{\mathcal{Z}_t(y;\gamma)}\partial_t p_t(z) + p_t(z)\partial_t\left[\frac{p_t^\gamma(y|z)}{\mathcal{Z}_t(y;\gamma)}\right]$$

$$= \partial_t p_t^{(\gamma)}(z|y).$$

Therefore, in this case, we cannot sample from $p_1^{(\gamma)}$ with the transition rate $u_t^{(\gamma)}$; for empirical results, see the simulations in Section 4.1. When $\gamma = 1$, the Kolmogorov forward equation holds, since

$$p_t(z)\partial_t\left[\frac{p_t^\gamma(y|z)}{\mathcal{Z}_t(y;\gamma)}\right] = p_t(z)\partial_t\left[\frac{p_t(z|y)}{p_t(z)}\right]$$

$$= \partial_t p_t(z|y) - p_t(z|y)\frac{\partial_t p_t(z)}{p_t(z)}$$

$$= -\sum_x Q_t(z, x)p_t(x|y) + \frac{p_t(z|y)}{p_t(z)}\sum_x Q_t(z, x)p_t(x)$$

$$= -\sum_{x\neq z}\left[Q_t(z, x)p_t(x|y) - \frac{p_t(z|y)}{p_t(z)}Q_t(z, x)p_t(x)\right]$$

$$= p_t(z)\sum_{x\neq z} u_t(x, z)\frac{[p_t(y|z) - p_t(y|x)]}{p(y)},$$

where $Q_t$ is the transition rate that can generate $p_t(x|y)$ and $p_t(x)$ from $t = 1$ to $t = 0$.

In contrast, as demonstrated in Theorem 2, the rate-based guidance

$$u_t^{(\gamma)}(z, x|y) = \left[\frac{\mathbb{E}_{p_{1|t}(\mathbf{x}_1|z)}p_1^\gamma(y|\mathbf{x}_1)}{\mathbb{E}_{p_{1|t}(\mathbf{x}_1|x)}p_1^\gamma(y|\mathbf{x}_1)}\right]u_t(z, x) \tag{19}$$

allows us to sample from the guided distribution $p_1^{(\gamma)}$ with the probability path $p_t^{(\star,\gamma)}(x|y) \propto \sum_{x_1} p_1(x_1)p_1^\gamma(y|x_1)p_{t|1}(x|x_1)$. The difference between the proposed rate-based guidance (equation 19) and the predictor guidance (equation 18) in discrete state space is similar to that between the exact guidance and the classifier guidance in continuous state space; see Lemma 4.10 of Zhang et al. (2025).

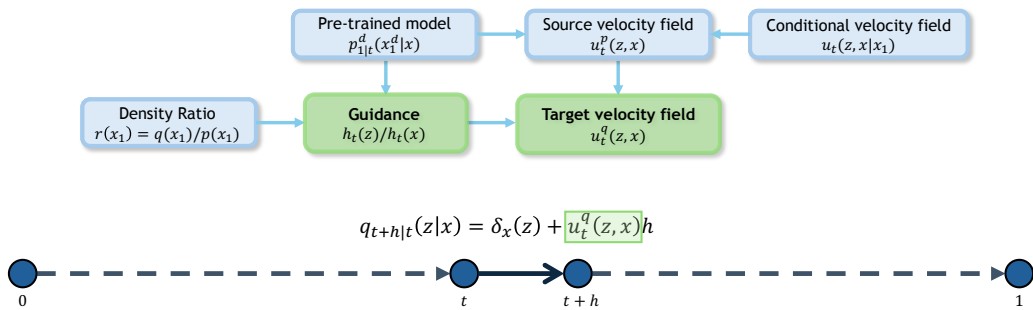

Figure 4: Framework of the improved rate-based guidance on discrete flow matching.

# F TRAINING AND SAMPLING FOR RATE-BASED GUIDANCE

## F.1 TRAINING OBJECTIVE

The rate-based guidance is the following scaler-valued function

$$h_t(x) \triangleq \mathbb{E}_{\mathbf{x}_1 \sim p_{1|t}(\mathbf{x}_1|x)}[r(\mathbf{x}_1)] = \sum_{s \in \mathcal{S}} h_t^d(s, x) p_{1|t}^d(s, x), \text{ for any } d \in [\mathcal{D}],$$

which is also the minimizer of the following objective:

$$\mathcal{L}_{h,p}^{\text{rate}}(\theta) = \mathbb{E}_{t \sim \mathcal{U}([0,1]), \mathbf{x}_1 \sim p_1(\mathbf{x}_1), \mathbf{x}_t \sim p_{t|1}(\mathbf{x}_t|\mathbf{x}_1)} \left[ h_t^\theta(\mathbf{x}_t) - r(\mathbf{x}_1) \log h_t^\theta(\mathbf{x}_t) \right]. \tag{20}$$

In practice, we can directly use this training objective to avoid additional summation operations.

## F.2 SAMPLING

We can construct a new conditional quantity $\check{u}_t^{q,d}(z^d, \mathbf{x}_t^d | \mathbf{x}_1^d) = \frac{h_t^\theta(z)}{h_t^\theta(\mathbf{x}_t)} u_t^p(z^d, \mathbf{x}_t^d | \mathbf{x}_1^d)$ for $z^d \neq \mathbf{x}_t^d$, $z^{\backslash d} = \mathbf{x}^{\backslash d}$ and $\check{u}_t^{q,d}(\mathbf{x}_t^d, \mathbf{x}_t^d | \mathbf{x}_1^d) = -\sum_{z^d \neq \mathbf{x}_t^d} \check{u}_t^{q,d}(z^d, \mathbf{x}_t^d | \mathbf{x}_1^d)$, and then use it instead of $u_t^{q,d}$ in equation 14. The sampling algorithm can be found in Algorithm 3, and the overall framework is illustrated in Fig. 4.

# G APPLICATION TO THE RL STAGE IN RLHF

## G.1 BACKGROUND AND MOTIVATION

In the diffusion language models (e.g. LLaDa, Nie et al., 2025c) and flow-based language models (e.g. Fudoki, Wang et al., 2025), we can obtain the pretrained model. Recently, existing works have developed post-training methods to improve preference through DPO (Rafailov et al., 2023) or RLHF (Ouyang et al., 2022). The main challenge is that the joint posterior predictor cannot be easily decoupled like AR models. To overcome this issue, some approximation strategies have been proposed for estimating log probability, such as empirical ELBO approximation (Zhu et al., 2025) and mean-field approximation (Zhao et al., 2025; Yang et al., 2025). However, these methods might introduce large approximation errors when the length of the sentence is large. Fortunately, our discrete guidance can provide a novel framework for alignment of diffusion and flow-based language models without approximation.

## G.2 ROADMAP OF ALIGNMENT FOR FLOW-BASED LANGUAGE MODELS

**Designing conditional path and rate.** Similar to Wang et al. (2025), in practice, we take metric-induced probability path with kinetic-optimal conditional transition rate Shaul et al. (2025):

$$p_{t|1}(\mathbf{o}_t^d | \mathbf{o}_1^d) = q_{t|1}(\mathbf{o}_t^d | \mathbf{o}_1^d) = \text{softmax}(-\beta_t d(\mathbf{o}_t^d, \mathbf{o}_1^d));$$
$$u_t^{p,d}(z^d, x^d | \mathbf{o}_1^d) = u_t^{q,d}(z^d, x^d | \mathbf{o}_1^d) = p_{t|1}(x | \mathbf{o}_1^d) \dot{\beta}_t [d(x^d, \mathbf{o}_1^d) - d(z^d, \mathbf{o}_1^d)]_+ \text{ for } z^d \neq x^d, \tag{21}$$

---

**Algorithm 3** Sampling with Rate-Based Guidance

---

**Require:** pretrained posterior $p_{1|t}$, conditional transition rate $u_t^p(z, x|x_1)$, initial value $x_0$, rate-based guidance $h_t^\theta$, step size $h$

1: $t \leftarrow 0$
2: $\mathbf{x}_t \leftarrow x_0$
3: **while** $t + h < 1$ **do**
4:      **for** $d = 1, \ldots, \mathcal{D}$ **do**                                  ▷ in parallel
5:          Sample $\mathbf{x}_1^d \sim p_{1|t}^d(\cdot \mid \mathbf{x}_t)$
6:          Calculate $\hat{u}_t^{d,\theta}(s, \mathbf{x}_t^d | \mathbf{x}_1^d) = \dfrac{h_t^\theta(\mathbf{x}_t^1, \ldots, \mathbf{x}_t^{d-1}, s, \mathbf{x}_t^{d+1}, \ldots, \mathbf{x}_t^{\mathcal{D}})}{h_t^\theta(\mathbf{x}_t)} u_t^{p,d}(s, \mathbf{x}_t^d | \mathbf{x}_1^d), s \neq \mathbf{x}_t^d$
7:          $\lambda^d \leftarrow \sum_{s \neq \mathbf{x}_t^d} \hat{u}_t^{d,\theta}(s, \mathbf{x}_t^d | \mathbf{x}_1^d)$
8:          Sample $Z_{\text{jump}}^d \sim U[0, 1]$
9:          **if** $Z_{\text{jump}}^d \leq 1 - e^{-h\lambda^d}$ **then**
10:              Sample $\mathbf{x}_t^d \sim \dfrac{\hat{u}_t^{d,\theta}(\cdot, \mathbf{x}_t^d | \mathbf{x}_1^d)}{\lambda^d} \left(1 - \delta_{\mathbf{x}_t^d}(\cdot)\right)$
11:          **end if**
12:      **end for**
13:      $t \leftarrow t + h$
14: **end while**
15: $t \leftarrow t - h$
16: **for** $d = 1, \ldots, \mathcal{D}$ **do**                                       ▷ in parallel
17:      Sample $\mathbf{x}_1^d \sim p_{1|t}^{d,\theta}(\cdot \mid \mathbf{x}_t)$
18: **end for**
19: **return** $\mathbf{x}_1$

---

where $\beta_t$ is an increasing function of $t$ with $\beta_0 = 0, \beta_1 = \infty$, and $d(\cdot, \cdot)$ is a metric.

**Training guidance model.** We can simply extend the unconditional training objective in the previous section to the following conditional version:

$$\mathcal{L}_{h,\pi_{ref}}(\theta) + \lambda \mathcal{L}_{h,\pi^*}(\theta)$$

$$= \mathbb{E}_{t \sim \mathcal{U}([0,1]), \mathbf{c} \sim p_{\mathbf{c}}, \mathbf{o}_1 | \mathbf{c} \sim \pi_{ref}, \mathbf{o}_t | \mathbf{o}_1 \sim p_{t|1}} \Big[ \sum_{d=1}^{\mathcal{D}} h_t^{d,\theta}(\mathbf{o}_1^d, \mathbf{o}_t, \mathbf{c}) - \exp\left(\frac{\mathcal{R}(\mathbf{c}, \mathbf{o}_1)}{\tau}\right) \log h_t^{d,\theta}(\mathbf{o}_1^d, \mathbf{o}_t, \mathbf{c}) \Big]$$

$$+ \lambda \mathbb{E}_{t \sim \mathcal{U}([0,1]), \mathbf{c} \sim p_{\mathbf{c}}, \mathbf{o}_1 | \mathbf{c} \sim \pi^*, \mathbf{o}_t | \mathbf{o}_1 \sim q_{t|1}} \Big[ \sum_{d=1}^{\mathcal{D}} \log \Big( \sum_{s \in \mathcal{S}} h_t^{d,\theta}(s, \mathbf{o}_t, \mathbf{c}) p_{1|t}^d(s | \mathbf{c}, \mathbf{o}_t) \Big) - \log h_t^{d,\theta}(\mathbf{o}_1^d, \mathbf{o}_t, \mathbf{c}) \Big].$$

$$(22)$$

**Sampling.** Given the prompt $\mathbf{c}$, at the current time $t$, for each $d \in [\mathcal{D}]$, we first sample $\mathbf{o}_1^d$ from

$$q_{1|t}^{d,\theta}(\mathbf{o}_1^d | \mathbf{c}, \mathbf{o}_t) \propto h_t^{d,\theta}(\mathbf{o}_1^d, \mathbf{o}_t, \mathbf{c}) p_{1|t}^d(\mathbf{o}_1^d | \mathbf{c}, \mathbf{o}_t),$$

and then sample $\mathbf{o}_{t+h}^d$ similar to equation 14.

# H    ADDITIONAL RESULTS

## H.1    IMPLEMENTATION DETAILS OF ENERGY-GUIDED SAMPLING

For a fair comparison, we train the guidance models (conditional expectation) using the Bregman divergence (Equation 11 and Equation 20) without regularization. For the rate-based guidance and predictor guidance, we take the initial distribution $p_0 = p_0^{(\gamma)} = \delta_{\mathrm{m}}$ for efficient sampling, where $\mathrm{m}$ is the mask state. For sampling with the posterior-based guidance, we consider both masked and uniform initial distributions. We used the mixture probability path and the associated transition rate (Equation 4) with the cosine time schedule $\kappa_t = \cos^2\left[\frac{\pi}{2}(1 - t)\right]$ in our 2-D experiments, which is kinetic optimal as mentioned in Shaul et al. (2025).

Our pretrained model and guidance models are SiLU networks with 3 hidden layers with dimension 256. For training, we use 100,000 datapoints from source data distribution. The optimizer is Adam with learning rate 1e-4.

## H.2 IMPLEMENTATION DETAILS OF MULTIMODAL TASKS

To match the conditional probability path used in Wang et al. (2025), we use metric-induced path with metric $d(f(x), f(z)) = \|\tilde{x} - \tilde{z}\|^4$, where $\tilde{x}, \tilde{z}$ are normalized token embeddings which are taken from the original text embedding layer of Janus-Pro-7B and the image embeddings of Llama-Gen; for time schedule $\beta_t$, we set $\beta_t = 3(\frac{t}{1-t})^{0.9}$ as suggested in Shaul et al. (2025).

Our guidance model is a network composed of 6 LLaDA blocks (Nie et al., 2025b) with hidden dimension 768, a time embedding mapping $\mathbb{R}$ to a hidden dimension of 768, a token embedding mapping each token into a 768-dimensional space.

## H.3 EFFECTIVENESS OF REGULARIZATION IN A 2-D EXPERIMENT

In this subsection, we examine the effectiveness of the regularization term in the training objective. Samples of the source and target distributions are shown in Fig. 5, and the two distributions differ substantially. We first pretrain a posterior model on the source data, subsequently pretrain a density ratio model by minimizing the following training objective:

$$\mathcal{L}_r(\theta) = -\mathbb{E}_{\mathbf{x}_1 \sim p_1(\mathbf{x}_1)} \log \left( \frac{1}{r^\theta(\mathbf{x}_1) + 1} \right) - \mathbb{E}_{\mathbf{x}_1 \sim q_1(\mathbf{x}_1)} \log \left( \frac{r^\theta(\mathbf{x}_1)}{r^\theta(\mathbf{x}_1) + 1} \right).$$

We then train posterior-based guidance with the proposed training objective $\mathcal{L}_{h,p}(\theta) + \lambda \mathcal{L}_{h,q}(\theta)$, sweeping $\lambda$ from 0 to 1 in steps of 0.2. Here, we take the uniform initial distribution. Finally, we generate samples using Algorithm 2. As presented in Fig. 6, the posterior-based guidance trained with a large hyperparameter offers a distribution closer to the target distribution.

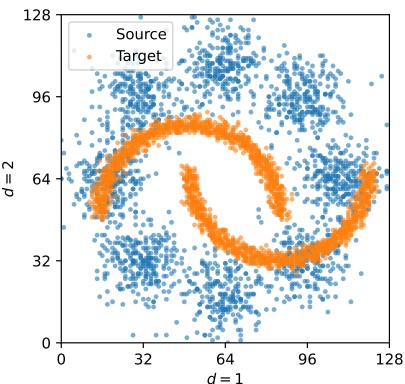

Figure 5: Some samples of source and target distributions.

## H.4 ADDITIONAL 2-D RESULTS

Additional results on 2-D simulations are provided in Fig. 7, Fig. 8, Fig. 9, Fig. 10, Fig. 11, and Fig. 12.

## H.5 ABLATION STUDY OF THE REGULARIZATION STRENGTH

In Appendix H.5, we present an ablation study on the effect of the regularization strength introduced in Section 3.3.

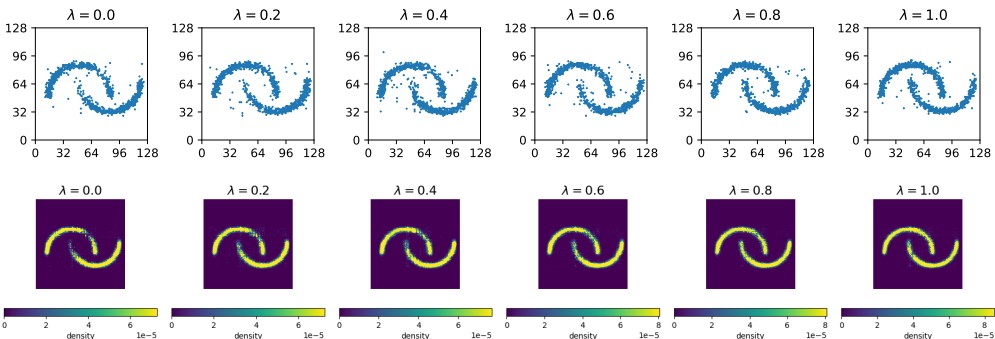

Figure 6: First row: samples generated by posterior-based guidance (64 sampling steps) with different hyper-parameters in the training objective. The initial distribution is uniform. Second row: density estimation of the target distribution using posterior-based guidance with different hyperparameters.

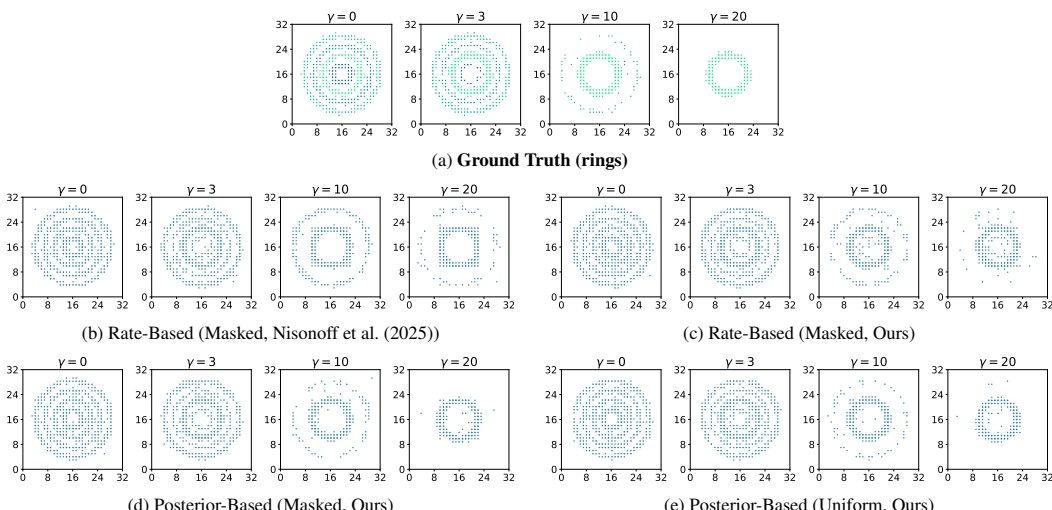

Figure 7: Comparison of sampling results with different guidance schemes (64 sampling steps) in 2-D experiments. (a) The samples of the guided distribution $p_1^{(\gamma)}$ with different guidance strength $\gamma$; (b) the data sampled by the predictor guidance (Nisonoff et al., 2025) with masked initial distribution; (c) the data sampled by the proposed rate-based guidance with masked initial distribution; (d) the data sampled by the proposed posterior-based guidance with masked initial distribution; (e) the data sampled by the proposed posterior-based guidance with uniform initial distribution.

Table 5: Ablation study of the regularization strength.

| Method | Single Obj. | Two Obj. | Counting | Colors | Position | Color Attri. | Overall ↑ |
|---|---|---|---|---|---|---|---|
| FUDOKI | 96.25 | 83.84 | 47.50 | 91.49 | 71.00 | 74.00 | 77.35 |
| $\eta = 0.1$ | 96.25 | 90.91 | 45.00 | 91.49 | 68.00 | 71.00 | 77.11 |
| $\eta = 0.5$ | 93.75 | 85.86 | 52.50 | 89.36 | 70.00 | 77.00 | 78.08 |
| $\eta = 1.0$ | 92.50 | 87.88 | 52.50 | 92.55 | 67.00 | 74.00 | 77.74 |

*Note:* Results are percentages.

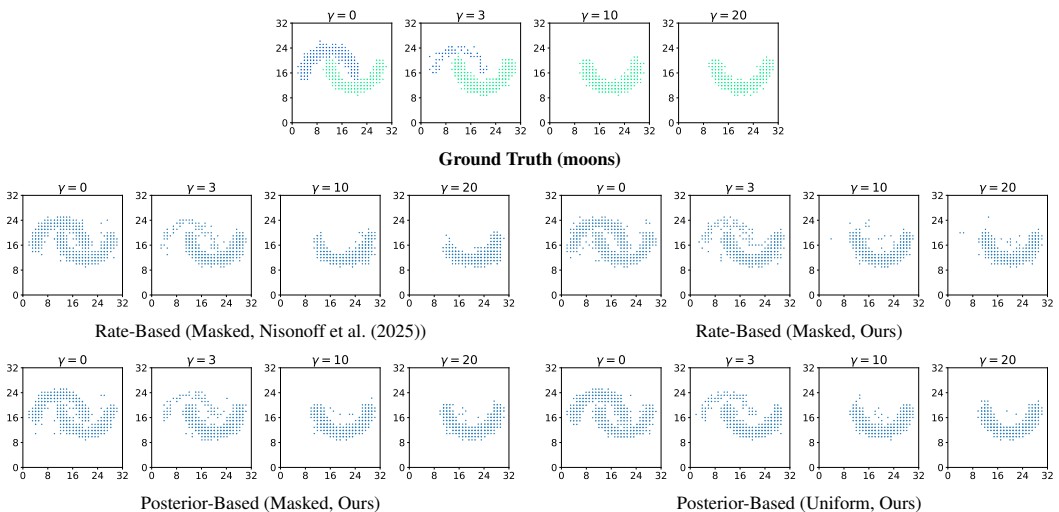

Figure 8: Comparison of sampling results with different guidance schemes in moons.

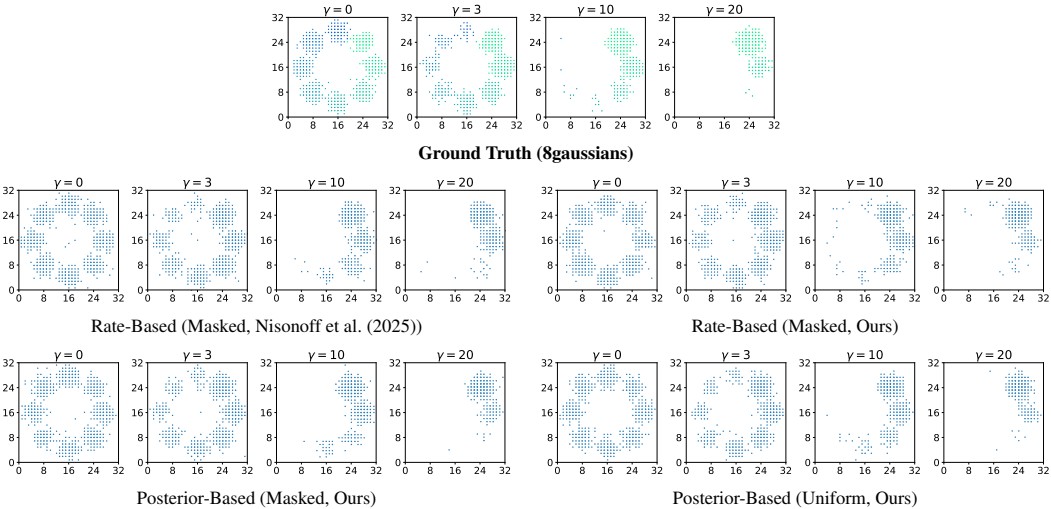

Figure 9: Comparison of sampling results with different guidance schemes in 8gaussians.

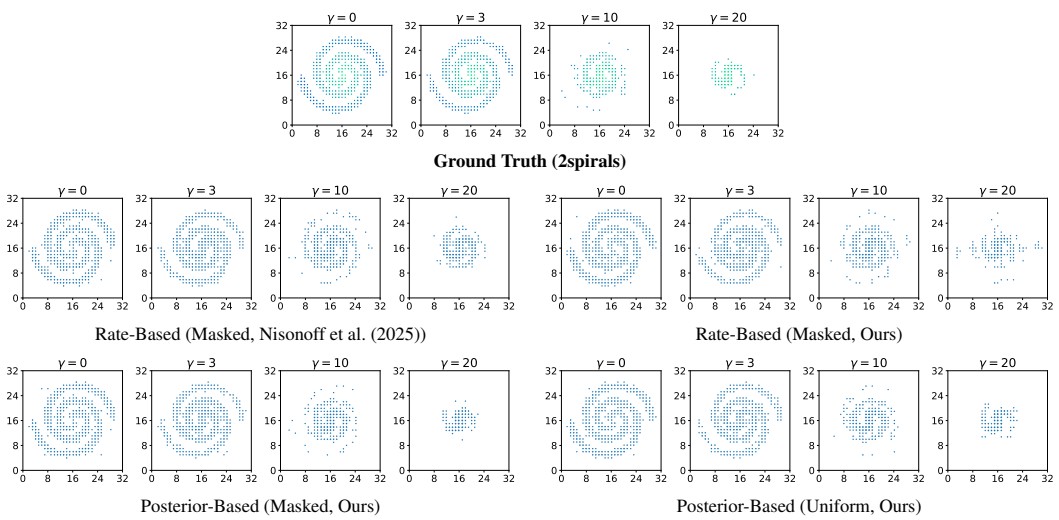

Figure 10: Comparison of sampling results with different guidance schemes in 2spirals.

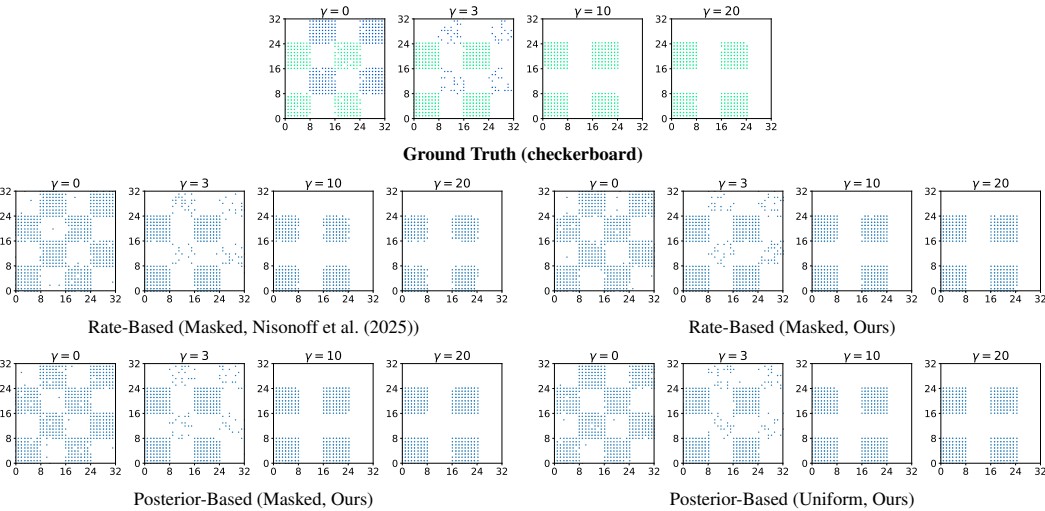

Figure 11: Comparison of sampling results with different guidance schemes in Checkerboard.

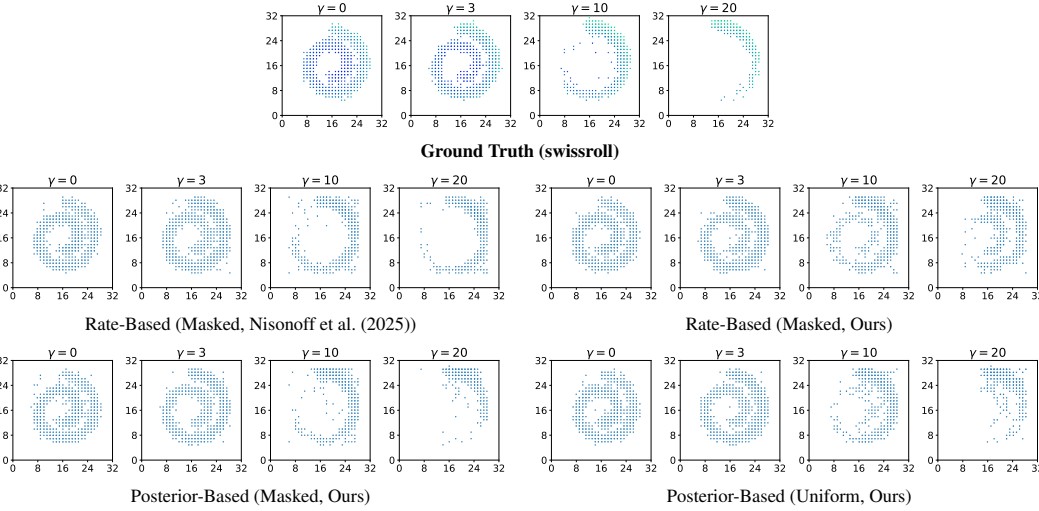

Figure 12: Comparison of sampling results with different guidance schemes in Swissroll.

## LLM USAGE

In our multimodal understanding experiments, we follow the standard practice of using LLM-as-a-judge to assess rewards and evaluations based on the question, ground-truth answer, and model response. LLMs are further used solely for polishing the writing. Importantly, no novel ideas, analyses, or discoveries are contributed by LLMs.

