# OpenReview forum: "Discrete Guidance Matching: Exact Guidance for Discrete Flow Matching"
_ICLR.cc/2026/Conference — ICLR 2026 Poster_

### Official Review · Reviewer_nrvw · 2025-10-28

**Soundness:** 2
**Presentation:** 2
**Contribution:** 2
**Rating:** 2
**Confidence:** 3

**Summary:**

The paper introduces Discrete Guidance Matching (DGM), a posterior-based guidance rule intended for discrete flow and diffusion models. The central theoretical statement asserts that, provided the density ratio \(r(x)=q_1(x)/p_1(x)\) is known and the source and target conditional paths match, one can recover the target posterior by reweighting the source posterior at each step. The manuscript positions this identity as a unifying lens on class-conditional guidance, energy-guided sampling, and RLHF-like objectives, and argues that the resulting sampler uses a single forward pass per step rather than the multiple passes required by rate-based or first-order schemes. Empirically, the paper demonstrates a two-dimensional toy “energy-guidance” setup and applies the approach to a multimodal discrete flow model finetuned with rewards, reporting modest improvements on GenEval and a composite multimodal score. Despite the tidy narrative, the practical validity of the assumptions and the evidentiary weight of the experiments remain limited; taken together, the contribution feels incremental, and I recommend rejection.

**Strengths:**

Conceptually, the work is clean and easy to follow: the authors articulate a clear posterior-reweighting view that helps connect several strands of discrete guidance under a single umbrella. The derivations are presented with serviceable notation, and the core loss used to fit the guidance term—cast via a Bregman-style objective with an optional target-data regularizer—is straightforward to implement in existing codebases. The paper also does a decent job motivating why rate-based or gradient-first-order methods can be computationally heavy in discrete settings, and it frames the proposed posterior-based rule as a way to reduce the number of model calls at sampling time. The organization is generally strong, with definitions and assumptions stated upfront, and the training/evaluation pipeline, while narrow, is at least described with sufficient detail to enable reproduction by a motivated practitioner.

**Weaknesses:**

The primary weakness is that the advertised “exactness” depends on an oracle-like requirement—knowing or accurately estimating the terminal density ratio—and on heavy modeling constraints—absolute continuity \(q_1\!\ll\!p_1\) and equality of the conditional paths—that are rarely guaranteed in real discrete generative pipelines. There is no robustness theory that translates estimation error in \(\widehat r\) into bounds on divergence between the induced terminal law and the desired target, nor is there sensitivity analysis for violations of absolute continuity or path mismatch; as a result, the central theorem’s practical relevance is unclear. The empirical section compounds this concern: the compute narrative around “one forward pass per step” is not cleanly audited, with timing comparisons confounded by different initial distributions and no end-to-end latency or throughput reported on the main RLHF tasks, and the additional cost of training a ratio model plus guidance head is not measured. Methodologically, the work criticizes first-order approximations for discrete data yet employs a first-order update in its own sampler without analyzing the induced bias, which undercuts the rhetorical stance. The evaluation design risks reward leakage by incorporating evaluation metrics into the reward itself, while the reported gains are small, lack statistical characterization (no confidence intervals, multi-seed variance, or significance tests), and are restricted to a narrow set of settings; broader discrete domains (language-only, program synthesis, tabular) and standard discrete diffusions are not explored. Finally, the “target-data regularizer” raises circularity concerns in scenarios where access to \(x_1\sim q_1\) is precisely what guidance is meant to provide, and several strong baselines typical in modern RLHF or discrete-guidance literature are missing, making it difficult to assess how much the method truly advances the state of the art.

**Questions:**

The core theoretical promise of “exactness” rests on conditions that are rarely met in realistic discrete pipelines and on an oracle-like requirement that the terminal density ratio be known or accurately estimated; absent a robustness analysis, the practical force of the theorem is limited. In particular, the absolute-continuity requirement \(q_1\!\ll\!p_1\) can be violated in RLHF or reward-driven regimes where the target allocates mass to sequences/events that the base model assigns essentially zero probability, leading to degeneracy of importance weights and ill-posed guidance; the manuscript neither diagnoses these failure modes nor proposes remedies (support repair, smoothing, clipping) or theoretical guarantees under misspecification.

The conditional-path matching assumption is likewise heavy: equating \(p_{t|1}\) and \(q_{t|1}\) (often with matched corruption schedules) is nontrivial outside the authors’ favored flow parameterization, and there is no sensitivity or stability bound quantifying how a small path mismatch per step compounds into terminal bias. Even within the proposed posterior scheme, the sampler relies on a first-order update while simultaneously critiquing first-order approximations for discrete data; the resulting approximation error is unanalyzed and unablated, which undercuts the “exact” branding. On the empirical side, the efficiency narrative is not audited end-to-end: the 2-D timing study confounds initialization (masked vs. uniform) and does not report system-level latency/throughput for the main tasks under fixed hardware, batch, and backend, while the additional training cost for the ratio estimator and guidance head is omitted—function-call bookkeeping is not a substitute for wall-clock and GPU-hour accounting. The evaluation design further risks reward leakage by optimizing with a metric (e.g., GenEval) used again at test time, blurring the line between genuine generalization and metric overfitting; moreover, reported gains are small, lack multi-seed variance or confidence intervals, and do not survive scrutiny without statistical characterization. Breadth is limited: beyond a toy 2-D setup and a single multimodal flow, there is no coverage of standard discrete diffusion baselines (e.g., multinomial/discrete diffusion, argmax flows) or of diverse discrete domains such as language-only generation, program synthesis, or tabular/graph settings; critical ablations are missing (quality of the learned \(r\), robustness to support violations, influence of the step-size/first-order update, and sensitivity to the initial distribution). Finally, the “target-data regularizer” presumes access to \(x_1\sim q_1\), which is conceptually circular when the whole point of guidance is to approximate \(q_1\) in the first place.

These issues are compounded by substantial missing baselines across discrete diffusion/guidance and RLHF preference optimization: beyond a couple of handpicked methods, the paper omits widely used, directly comparable approaches in both families. Representative missing or under-engaged baselines that should be reported under matched hardware, batch size, and privacy/backend settings include Dhariwal & Nichol (2021, classifier guidance for diffusion), Ho & Salimans (2022, classifier-free guidance), Austin et al. (2021, D3PM for discrete state spaces), Hoogeboom et al. (2021, autoregressive/discrete diffusion), Chang et al. (2022, MaskGIT as a discrete generative baseline), Ouyang et al. (2022, RLHF for instruction following as a canonical PPO/SFT+RM baseline), Rafailov et al. (2023, DPO as a strong non-RL preference optimizer), Yuan et al. (2023, RRHF), Xu et al. (2023, ImageReward as a modern reward model for T2I), Kirstain et al. (2023, PickScore as a learned preference scorer), Wu et al. (2024, HPS v2 as a unified human-preference score), Dathathri et al. (2020, PPLM for posterior control of discrete text, closely related in spirit to guidance), and Bai et al. (2022, Constitutional AI / RLAIF variants showing strong preference optimization without explicit ratio models). Including these methods would stress-test the claimed unification and computational advantage, reveal where DGM sits relative to best-practice discrete diffusion/guidance and modern preference optimization, and help separate genuine algorithmic gains from metric leakage or implementation choices.

**missing references:**
Dhariwal, P., & Nichol, A. (2021). Diffusion Models Beat GANs on Image Synthesis. *NeurIPS.*
Ho, J., & Salimans, T. (2022). Classifier-Free Diffusion Guidance. *arXiv:2207.12598.*
Austin, J., et al. (2021). Structured Denoising Diffusion Models in Discrete State-Spaces (D3PM). *NeurIPS.*
Hoogeboom, E., et al. (2021). Autoregressive Discrete Diffusion. *arXiv:2103.12027.*
Chang, W., et al. (2022). MaskGIT: Masked Generative Image Transformer. *CVPR.*
Ouyang, L., et al. (2022). Training Language Models to Follow Instructions with Human Feedback. *NeurIPS.*
Rafailov, R., et al. (2023). Direct Preference Optimization: Your Language Model is Secretly a Reward Model. *NeurIPS.*
Yuan, W., et al. (2023). RRHF: Rank Responses to Align Language Models with Human Feedback. *NeurIPS.*
Xu, K., et al. (2023). ImageReward: Learning and Evaluating Human Preferences for Text-to-Image Generation. *NeurIPS.*
Kirstain, Y., et al. (2023). PickScore: A Large-Scale Human Preference Dataset and Scorer for Text-to-Image. *arXiv.*
Wu, J., et al. (2024). HPS v2: A Unified Human Preference Score for Text-to-Image Evaluation. *arXiv.*
Dathathri, S., et al. (2020). Plug-and-Play Language Models: A Simple Approach to Controlled Text Generation. *ICLR.*
Bai, Y., et al. (2022). Constitutional AI: Harmlessness from AI Feedback. *arXiv.*

**Details Of Ethics Concerns:**

Can you provide a robustness result that quantitatively links error in the learned density ratio \(\widehat r\) (e.g., in sup norm or an f-divergence) and any violations of absolute continuity or path matching to a provable bound on the gap between the sampler’s terminal distribution and \(q_1\), thereby clarifying when “exactness” meaningfully approximates reality? What precisely constitutes the claimed “single forward pass” at sampling time—do you count the base posterior model and the guidance head—and can you report end-to-end latency and throughput on your RLHF tasks under a fixed hardware, batch size, and backend, alongside a corrected two-dimensional timing study that uses identical initializations across methods? Given that your reward includes evaluation-aligned metrics (e.g., GenEval), can you ablate them from the reward or add held-out metrics not used for training to demonstrate genuine generalization and avoid metric leakage, and can you expand the baseline set to include stronger contemporary RLHF/discrete-guidance approaches? Finally, since your sampler still applies a first-order update despite arguing such approximations are problematic in discrete spaces, can you analyze the resulting approximation error and show that your empirical gains persist under alternative update schemes or step-size regimes so that conclusions do not hinge on this choice?

---

### Official Review · Reviewer_N6g7 · 2025-10-30

**Soundness:** 2
**Presentation:** 3
**Contribution:** 3
**Rating:** 6
**Confidence:** 3

**Summary:**

The paper proposes Discrete Guidance Matching, a CTMC-based framework for exact, efficient guidance in discrete generative models like diffusion and flow matching. It derives precise transition rates via density ratio correction, requiring one model pass per step. Training minimizes Bregman divergence with regularization. It generalizes prior methods and outperforms baselines in tasks like energy-guided sampling, text-to-image preference alignment, and multimodal understanding, with faster sampling.

**Strengths:**

1. Unlike existing methods relying on first-order Taylor approximation, the paper derives the exact transition rate for the target distribution using density ratios, eliminating non-negligible approximation errors in discrete state spaces.
2. The framework encompasses existing guidance methods, such as class-conditional and energy-weighted guidance, as special cases and seamlessly applies to both discrete flow models and discrete diffusion models, including masked diffusion.
3. Each sampling step requires only 1 forward pass vs. multiple function calls in rate-based or approximated methods, leading to faster sampling than rate-based baselines.

**Weaknesses:**

1. Theoretical Soundness Gaps in CTMC: The paper fails to verify that the derived target transition rate satisfies CTMC’s core conservativeness condition $ u_t^q(x,x) = -\sum_{z \neq x} u_t^q(z,x) $. This omission means the transition rate may not form a valid CTMC, breaking the probabilistic consistency of the generation process.
2. Unsubstantiated High-Dimensional Assumption: The coordinate-wise independent transition rate assumption, like for high-dimensional data, contradicts the inherent dimensional coupling in real-world discrete tasks such as long text and high-res images. The paper only validates its performance in 2D low-dimensional spaces and provides no analysis of high-dimensional performance, thereby limiting its practical utility.
3. Missing Non-Negativity Constraints for Density Ratios: The Bregman divergence training with $ F(x)=<x, log x> $ does not guarantee non-negative density ratios $ r(x)=q_1(x)/p_1(x) $. Negative density ratios would lead to "negative probability transitions," which violate CTMC physics; however, the paper lacks regularization or pre-sampling checks to prevent this.
4. Incomplete Experimental Scope: High-dimensional tasks such as 32×32 discrete images and low-target-sample regimes, which are critical for real-world few-shot alignment, are not tested, leaving key practical questions unanswered.

**Questions:**

1. Did you verify that the derived target transition rate satisfies CTMC’s conservativeness condition $ u_t^q(x,x) = -\sum_{z \neq x} u_t^q(z,x) $? If yes, please include the proof in the revised manuscript; if not, how do you address the resulting probabilistic inconsistency?
2. The coordinate-wise independence assumption for high-dimensional data ignores dimensional coupling. Do you have a strategy to model this coupling, for example, low-rank decomposition, or pairwise terms? Can you provide experimental results on high-dimensional tasks like $ D=512 $ text sequences?
3. How do you ensure the estimated density ratio $ r(x) $ remains non-negative during training and sampling? Have you observed negative density ratios in experiments, and if so, how did you resolve them?
4. For low-target-sample regimes like 10-shot or 1-shot preference alignment, does your regularization term still work? If not, do you have a weak-supervision or prior-based alternative to improve data efficiency?

---

> ### Author Response · Authors · 2025-11-21
>
> We appreciate the reviewer’s helpful comments and the time invested in assessing our work. The feedback has been instrumental in refining both the presentation and the technical aspects of the paper.
>
> **Q**: *Theoretical Soundness Gaps in CTMC: The paper fails to verify that the derived target transition rate satisfies CTMC’s core conservativeness condition $u_t^q(x,x)=-\sum_{z\neq x}u_t^q(z,x)$. This omission means the transition rate may not form a valid CTMC, breaking the probabilistic consistency of the generation process.*
>
> **A**:  Thank you very much for your question. In the following, we verify that the transition rate derived by our posterior-based guidance satisfies both the rate property and the Kolmogorov forward equation. The key takeaway is that our theorem is the exact guidance for discrete flow matching, which means the derived form is the exact transition probability for the target distribution. Thus, it satisfies both the rate property and the Kolmogorov forward equation.
> 1. Rate property:
> \begin{align}
> \sum _z u_t^q(z,x)
> &= \sum_z \sum _{d} \delta _{x ^{\backslash d}}(z^{\backslash d}) u_t ^q(z^d, x) \\\\
> &= \sum _z \sum _{d} \delta _{x ^{\backslash d}}(z ^{\backslash d})
>     \sum _{x_1^d} u _t(z^d, x^d \mid x_1^d)
>     \frac{\mathbb{E} _{\mathbf{x} ^{\backslash d}_1 \sim p(\mathbf{x} ^{\backslash d}_1 \mid \mathbf{x} ^d_1 = x _1^d, \mathbf{x} _t = x)} \big[ r(\mathbf{x}_1) \big]}{\mathbb{E} _{\mathbf{x} _1 \sim p _{1|t}(\mathbf{x} _1 \mid x)} \big[ r(\mathbf{x} _1) \big]}
>     p _{1|t}(x _1^d \mid x) \\\\
> &= \sum _{d} \sum _{z^{\backslash d}} \delta _{x^{\backslash d}}(z^{\backslash d})
>     \sum _{x_1^d}
>     \Big[ \underbrace{\sum _{z^d} u _t(z^d, x^d \mid x_1^d)} _{=0} \Big]
>     \frac{\mathbb{E} _{\mathbf{x} ^{\backslash d} _1 \sim p(\mathbf{x}^{\backslash d}_1 \mid \mathbf{x}^d_1 = x_1^d, \mathbf{x}_t = x)} \Big[ r(\mathbf{x} _1) \Big]}{\mathbb{E} _{\mathbf{x} _1 \sim p _{1|t}(\mathbf{x} _1 \mid x)} \Big[ r(\mathbf{x} _1) \Big]}
>     p _{1|t}(x _1^d \mid x) \\\\
> &= 0.
> \end{align}
> where we use the rate property of the conditional transition rate in the last equation.
>
> 2. Kolmogorov forward equation:
> \begin{align}
>     \sum _{x}u _t ^q(z,x)q _t(x)=&~\sum _{x}\sum _{d}\delta _{x ^{\backslash d}}(z ^{\backslash d})\sum _{x_1 ^d}u _t(z ^d,x ^d|x _1^d)q _{1|t}(x ^d _1|x)q _t(x)\\\\
>     =& \sum _{x} \sum _{d} \sum _{x _1} \delta _{x ^{\backslash d}}(z ^{\backslash d})u _t(z ^d,x ^d | x _1^d)q _{1 | t}(x _1|x)q _t(x)\\\\
>     =& \sum _{x _1}\sum _{x}\sum _{d}\delta _{x^{\backslash d}}(z^{\backslash d})u _t(z^d,x^d|x _1^d)p _{t|1}(x|x _1)q _1(x _1)\\\\
>  =& \sum _{x _1}\dot{p} _{t|1}(x|x _1)q _1(x _1)\\\\
> =& \dot{q} _t(x),
> \end{align}
> where we use the Kolmogorov forward equation for the conditional probability path in the fourth equation.

---

> > ### Author Response · Authors · 2025-11-21
> >
> > **Q**: *Unsubstantiated High-Dimensional Assumption: The coordinate-wise independent transition rate assumption, like for high-dimensional data, contradicts the inherent dimensional coupling in real-world discrete tasks such as long text and high-res images. The paper only validates its performance in 2D low-dimensional spaces and provides no analysis of high-dimensional performance, thereby limiting its practical utility. The coordinate-wise independence assumption for high-dimensional data ignores dimensional coupling. Do you have a strategy to model this coupling, for example, low-rank decomposition, or pairwise terms? Can you provide experimental results on high-dimensional tasks like $D=512$ text sequences?*
> >
> > **A**:  Thank you very much for your question. Diffusion language model [1,2] and discrete flow matching [3,4,5,6] adopt the coordinate-wise independent transition rate matrix and achieve good performance on language, image, protein, and multimodal generation. As far as we know, there is no work be able to adopt coordinate-wise dependent transition rate matrix since the transition matrix would grow from $K \times K$ to $K^D \times K^D$, where $K$ denotes the vocabulary size and $D$ denotes the sequence length.
> >
> > In Section 4.1, we verify the effectiveness of our method through two-dimensional simulations. In Section 4.2, we verify the effectiveness of our method on text-to-image generation tasks and multimodal tasks, where the input and output sequence lengths $D=1076$.
> >
> > [1] Nie, S., Zhu, F., You, Z., Zhang, X., Ou, J., Hu, J., Zhou, J., Lin, Y., Wen, J., & Li, C. (2025). Large Language Diffusion Models. ICML.
> >
> > [2] Xin et al. (2025). Lumina-DiMOO: An Omni Diffusion Large Language Model for Multi-Modal Generation and Understanding. ArXiv, abs/2510.06308.
> >
> > [3] Campbell, A., Yim, J., Barzilay, R., Rainforth, T., & Jaakkola, T. (2024). Generative Flows on Discrete State-Spaces: Enabling Multimodal Flows with Applications to Protein Co-Design. ArXiv.
> >
> > [4] Wang, J., Lai, Y., Li, A., Zhang, S., Sun, J., Kang, N., Wu, C., Li, Z., & Luo, P. (2025). FUDOKI: Discrete Flow-based Unified Understanding and Generation via Kinetic-Optimal Velocities. ArXiv, abs/2505.20147.
> >
> > [5] Gat, I., Remez, T., Shaul, N., Kreuk, F., Chen, R.T., Synnaeve, G., Adi, Y., & Lipman, Y. (2024). Discrete Flow Matching. ArXiv, abs/2407.15595.
> >
> > [6] Luo, R., Xia, X., Wang, L., Chen, L., Shan, R., Luo, J., Yang, M., & Chua, T. (2025). NExT-OMNI: Towards Any-to-Any Omnimodal Foundation Models with Discrete Flow Matching. ArXiv, abs/2510.13721.
> >
> > **Q**: *Missing Non-Negativity Constraints for Density Ratios: The Bregman divergence training with $F(x)=<x, log x>$ does not guarantee non-negative density ratios $r(x)=q_1(x)/p_1(x)$. Negative density ratios would lead to "negative probability transitions," which violate CTMC physics; however, the paper lacks regularization or pre-sampling checks to prevent this. How do you ensure the estimated density ratio $r(x)$ remains non-negative during training and sampling? Have you observed negative density ratios in experiments, and if so, how did you resolve them?*
> >
> > **A**:  Thank you very much for your question. We would like to build a general guidance framework. Therefore, we use $q(x)$ to denote the target distribution without assuming any particular structure of the target distribution, and the density ratio $r(x)=q_1(x)/p_1(x)$ contains the sufficient and necessary information for guiding a pre-trained model on the source distribution to the target distribution. In real experiments, we verify the effectiveness of the proposed framework on conditional generation where $q_1(x)=p_1(x|y)=p_1(x)\frac{p(y|x)}{p(y)}$ in two-dimensional simulations and $q_1(x)=p_1(x) \frac{e^{\gamma r(x)}}{Z}$ in text-to-image generation tasks and multimodal understanding tasks. The ratio is predefined for each tasks, i.e., a pre-trained classifier for conditional generation and a pre-trained reward function. Therefore, we do not need to estimate the density ratio, which means we will not have negative probability transitions.

---

> > > ### Author Response · Authors · 2025-11-21
> > >
> > > **Q**: *Incomplete Experimental Scope: High-dimensional tasks such as 32×32 discrete images and low-target-sample regimes, which are critical for real-world few-shot alignment, are not tested, leaving key practical questions unanswered.*
> > >
> > > **A**:  Thank you very much for your question. In Section 4.2, we verify the effectiveness of the proposed framework on the current state-of-the-art multimodal discrete flow matching model. We improve the generation quality on text-to-image generation and multimodal understanding tasks. The resolution of the image is 384$\times$384.
> > >
> > >
> > > **Q**: *For low-target-sample regimes like 10-shot or 1-shot preference alignment, does your regularization term still work? If not, do you have a weak-supervision or prior-based alternative to improve data efficiency?*
> > >
> > > **A**:  Thank you very much for your question. In general, using information from the target distribution is helpful to learn the guidance network. However, if the number of samples from the target distribution is too small, it will induce the estimation error. Therefore, to alleviate the potential negative effect of the regularization, we need to choose the hyperparameter $\lambda$ (the strength of the regularization) carefully. Usually, it should be to $\sqrt{n/m}$, where $m$ and $n$ denote the number of samples in the source and target distribution, respectively.

---

> > > > ### Comment · Reviewer_N6g7 · 2025-11-24
> > > >
> > > > I thank the authors for their detailed response.
> > > >
> > > > The provided derivation regarding the CTMC conservativeness condition, the justification for the high-dimensional independence assumption, and the clarification on density ratio non-negativity have addressed my concerns.
> > > >
> > > > I have no further questions and maintain my positive rating.

---

> ### Comment · Reviewer_nrvw · 2025-11-24
> **Will you submit a rebutall to resolve my concerns?**
>
> Dear authors
>
> I note that rebuttals have been submitted to the other three reviewers, yet I have not received a response addressing my concerns. I kindly request the authors to provide a targeted rebuttal to address the points I raised. Should the rebuttal present reasonable and compelling arguments, I am willing to reconsider and update my review score accordingly.
>
> Reviewer nrvw

---

> ### Comment · Reviewer_N6g7 · 2025-11-24
>
> Hi Reviewer nrvw,
>
> I am just one of the reviewers. I noticed your recent comment on my slot asking for a rebuttal, and took a look at your initial review.
>
> I think the authors will be more than happy to provide the targeted rebuttal you are looking for. If you can kindly add an **official comment** in your review slot to clarify your demands, I recommend reframing your points into **crystal clear, actionable suggestions**, instead of asking general questions.
>
> Specifically, your points under the "Ethics Concerns" section currently read like general questions rather than specific ethical violations.
>
> Best,
>
> Reviewer N6g7

---

### Official Review · Reviewer_gFdm · 2025-10-31

**Soundness:** 3
**Presentation:** 2
**Contribution:** 3
**Rating:** 4
**Confidence:** 3

**Summary:**

This paper proposes a framework for general guidance in discrete flow models. While these models support unified understanding and generation for images and text, the framework works to improve the preference of the results according to specific reward signals.

**Strengths:**

1, The theory is solid. I am not a theory expert, but I validated the correctness of proof for theorem 1 and theorem 2.

2, The overall advantage in both generation and understanding proves the effectiveness of the proposed framework. From both quantitative results on T2I and case studies on 2D data, I can see considerable improvements compared to existing methods.

**Weaknesses:**

1, The presentation and delivery is problematic. The word Guidance should be used to describe inference-time guidance. When using an external reward to post-train the model, the method should be defined as RL (as stated in the Experiment Section). Also, the most referred reference in the paper [A] is a training-free guidance method for discrete flow matching model. If the authors want to categorize the method as guidance, at least they should explicitly notify the reason and the difference compared to existing guidance method (training is needed here).

Another critical problem in presentation is the missing of reasonable transition. For example, there is no clear problem definition on what your guidance wants to do in empirical tasks, and for theorem 1, no explanation is available on why we need to consider D-1 vectors in the discrete flow matching guidance (When I refer to your reference literature, they do not mention this D-1 term).

2, Due to the above misconception on RL and guidance, I am not sure whether it is suitable not to include other RL methods based on direct preference as baselines.

Reference:

[A] Unlocking Guidance for Discrete State-Space Diffusion and Flow Models

**Questions:**

1, Why do you consider your method with RL training as guidance?

2, Could you supplement the intuitive explanation for all your theorems and tell how they help to approach your method as a conclusion?

I would actively engage the rebuttal and would love to raise my score if my concerns are well solved.

---

> ### Author Response · Authors · 2025-11-21
>
> We thank the reviewer for the thoughtful comments and suggestions. We appreciate the time you spent on the paper and your recognition of our contributions and strengths. Below we address the concerns and comments that you have provided.
>
> **Q**: *The presentation and delivery is problematic. The word Guidance should be used to describe inference-time guidance. When using an external reward to post-train the model, the method should be defined as RL (as stated in the Experiment Section). Also, the most referred reference in the paper [A] is a training-free guidance method for discrete flow matching model. If the authors want to categorize the method as guidance, at least they should explicitly notify the reason and the difference compared to existing guidance method (training is needed here).*
>
> **A**:  Thank you very much for your question. We follow the same framework as the line of works on guidance[1,2, 3, 4, 5]. In general, guidance aims to provide a framework for posterior sampling by steering the generation process towards the desired distribution. Given the information of desired distribution $q_1(x)$, the guidance method aims to find out how to modify the existing pretrained model on source distribution $p_1(x)$. For energy-guided sampling [1,2,3], $q_1(x)=p_1(x) \frac{e^{-\gamma \mathcal{E}(x)}}{Z}$. The inverse problem/conditional generation [4, 5, 6], $q_1(x)=p_1(x|y)=p_1(x)\frac{p(y|x)}{p(y)}$, can be written as a special case of energy-guided sampling, where $\mathcal{E}(x)=-\log p(y=1|x)$. The guidance term can either be estimated by Monte Carlo estimation (training-free manner) [1, 4, 5, 6] or estimated by a trained neural network [1, 2, 3]. In our paper, we propose a general framework of guidance. In Section 4.1, we conduct experiments on inverse problem/conditional generation. In Section 4.2, we conduct experiments on RLHF setup, which belongs to a specific form of energy-guided sampling where $q_1(x)=p_1(x) \frac{e^{ r(x)/\tau}}{Z}$. This RLHF setup is also used in existing guidance work on continuous diffusion model [7].
>
>
> [1] Feng, R., Wu, T., Yu, C., Deng, W., & Hu, P. (2025). On the Guidance of Flow Matching. ICML.
>
> [2]Zhang, S., Zhang, W., & Gu, Q. (2025). Energy-Weighted Flow Matching for Offline Reinforcement Learning. ICLR.
>
> [3]Lu, C., Chen, H., Chen, J., Su, H., Li, C., & Zhu, J. (2023). Contrastive Energy Prediction for Exact Energy-Guided Diffusion Sampling in Offline Reinforcement Learning. ICML.
>
> [4]Song, J., Zhang, Q., Yin, H., Mardani, M., Liu, M., Kautz, J., Chen, Y., & Vahdat, A. (2023). Loss-Guided Diffusion Models for Plug-and-Play Controllable Generation. ICML.
>
> [5]Chung, H., Kim, J., McCann, M.T., Klasky, M.L., & Ye, J.C. (2023). Diffusion Posterior Sampling for General Noisy Inverse Problems. ICLR.
>
> [6]Nisonoff, H.M., Xiong, J., Allenspach, S., & Listgarten, J. (2025). Unlocking Guidance for Discrete State-Space Diffusion and Flow Models. ICLR.
>
> [7] Rector-Brooks, J., Hasan, M., Peng, Z., Quinn, Z., Liu, C., Mittal, S., Dziri, N., Bronstein, M.M., Bengio, Y., Chatterjee, P., Tong, A., & Bose, A.J. (2025). Steering Masked Discrete Diffusion Models via Discrete Denoising Posterior Prediction. ICLR.
>
> **Q**: *Another critical problem in presentation is the missing of reasonable transition. For example, there is no clear problem definition on what your guidance wants to do in empirical tasks, and for theorem 1, no explanation is available on why we need to consider D-1 vectors in the discrete flow matching guidance (When I refer to your reference literature, they do not mention this D-1 term).*
>
> **A**:  Thank you very much for your question. We aim to generate samples from a weighted distribution $q_1(x)\propto p_1(x)\exp(R(x)/\tau)$ in preference alignment tasks. We included a transition in Section 3.5 in the revised version. In Theorem 1, we derive the exact \textbf{transition probability} for the target distribution, while in Theorem 2, we derive the exact \textbf{transition rate} for the target distribution. Equation 2 in existing work [A] is a special case of Theorem 2 when the target distribution is a conditional distribution of the source distribution. The different properties of transition probability and transition rate are the main reasons we contain D-1 vectors in Theorem 1. When we use transition probability for inference, $q_{1|t}^d(x_1^d|x_t)$ is what the model used in practice. We can compute $q_{1|t}^d(x_1^d|x_t)$ by marginalized $p_{1|t}^d(x_1|x_t)$ over $x^{\backslash d}$) for different $d=1,2,\dots,D$.
>
> In general, our Theorem 1 is also valid for the transitional probability over D dimensions, i.e., in equation 15, we prove the $q _{1|t}(z|x)=\frac{r(z)}{\mathbb{E} _{\mathbf{x}_1\sim p _{1|t}(\mathbf{x} _1|x)}[r(\mathbf{x}_1)]}p _{1|t}(z|x)$.

---

> > ### Author Response · Authors · 2025-11-21
> >
> > **Q**: *Why do you consider your method with RL training as guidance? Due to the above misconception on RL and guidance, I am not sure whether it is suitable not to include other RL methods based on direct preference as baselines.*
> >
> > **A**:  Thank you very much for your question. RL training usually adopts policy gradient for updating the current policy network. Our paper works on steering a pre-trained generation process towards the desired distribution.
> > We only borrow the reward function to train a guidance network in Section 4.2, which is a special case for energy-based sampling. This setup is also used in existing guidance work on continuous diffusion model [7]. Moreover, we train the guidance network by minimizing the Bregman divergence rather than the policy gradient for the pre-trained model. Finally, RL algorithms use on-policy rollouts during training, which is more time-consuming than our supervised loss. Therefore, other RL methods may not be fair baselines with the guidance method.
> >
> > **Q**: *Could you supplement the intuitive explanation for all your theorems and tell how they help to approach your method as a conclusion?*
> >
> > **A**:  Thank you very much for your question. We derive the exact transition probability/rate of the target distribution given a pre-trained discrete flow matching model. The target transition probability of the target distribution is a weighted form of the pre-trained transition probability of the source distribution. The weighting term is a conditional expectation related to the transition probability in the source distribution $p_{1|t}$. In general, the optimal solution of a least-squares problem under the joint distribution recovers the conditional expectation we want. Therefore, we propose the corresponding training objective for calculating the weighting term.
> >
> > We found that our framework is general enough to deal with energy-based sampling and preference alignment tasks. Moreover, we found that its underlying theoretical tools are also general enough to derive guidance for continuous diffusion models, continuous flow matching, and discrete diffusion models, which is shown in Appendix E.

---

### Official Review · Reviewer_375z · 2025-10-31

**Soundness:** 3
**Presentation:** 3
**Contribution:** 3
**Rating:** 6
**Confidence:** 4

**Summary:**

Given a pre-trained discrete flow matching (DFM) model parametrized by a posterior $p_{1|t}$, and trained to generate samples from a distribution $p_1$ (somewhat confusing, it is denoted as the _source distribution_), the paper goal is to adapt the model to sample the target distribution $q_1=p_1(\cdot | y)$ where $y$ is a conditioning variable.

To achieve this, the authors propose to add ans "exact guidance" which learn an additional model $h_t (x_1^d, x_t)$ such that $$q_{1|t}(x_1^d|x_t) = \frac{h_t (x_1^d, x_t)}{\sum_s h_t (s, x_t)} p_{1|t}(x_1^d|x_t)$$ is the posterior of DFM model trained to sample the target distribution $q_1$.

Finally, the exact guidance method is evaluated mainly on the task of text-to-image.

**Strengths:**

1. The paper properly introduces its goal, the notation with a the required preliminaries work, and presents it method in a clear mathematical writing.

2. The proposed method attempts to advance fine tuning for conditional sampling which is a very important problem in generative models.

3. To the best of my knowledge the proposed method is innovative and constitutes a fair contribution.

**Weaknesses:**

While paper is interesting I still have a number of concerns:

1. The "RLHF ON MULTIMODAL BENCHMARK" subsection is lacking:

    a. The application of exact guidance to RLHF seems to be an important application of the method and should probably be introduced more clearly before the experiment section. In particular, explicitly state what object does $h_t^{\theta}$ approximate in the this settings. Additionally, even though the actual loss for training in this setting appears in the Appendix G, would good if there was a direct reference from the paper.

    b. In the main paper a single choice of probability path is introduced in equation 4. However going over the appendix it seems a different choice of probability path was used in the experiment, the metric induced path as in equation 21. Further, this probability path seems to use an additional metric which is not specified. While it is reasonable to leave these equations in the Appendix, it is expected to mention it in the experimental setup.

   c. Almost no implementation detail is provided for experiment: which pre-trained model is used is not clear, architecture and model size are not specified, and number of training iteration as well as effective batch size.

   d. It might be standard approach but I still find it disturbing that the GenEval reward model was used both in training and evaluation.

2. The sampling section is lacking, going over the appendix it seems there is a strong connection between previous suggested sampling algorithm and the exact guided sampling algorithm. Stressing the difference in the main paper could help the reader.

3. I find at least two mathematical claims that where introduced in the main paper but were not actually used by the proposed methods (negative claims):

    a. Theorem 2 is introduced and in the next three lines it is already stated that the authors do not make any use of it.

    b. Remark 1, this remark seems to claim again that the direct approximation of the rate would not yield the desired flow.

The paper is already heavy with alot of math to parse, and I would consider moving these to the appendix. Additionally, there is currently alot of important information in the appendix and it seems that rearranging the paper can possible make it alot easier to understand the main contribution and experiments of the paper.

**Questions:**

In Algorithm 2 Sampling with Posterior-Based Guidance it is state that the guided posterior needs to be calculated at each step, Since the sequence length and vocabulary size can potentially be large, how does this affect the sampling cost compared to the vanilla sampling in Algorithm 1?

---

> ### Author Response · Authors · 2025-11-21
>
> We sincerely thank the reviewer for the detailed and constructive feedback. We appreciate the careful reading of our manuscript and the insightful comments that have helped us significantly improve the clarity and scope of the revision.
>
> **Q**: *The application of exact guidance to RLHF seems to be an important application of the method and should probably be introduced more clearly before the experiment Section. In particular, explicitly state what object does $h_t^\theta$
>  approximate in the this settings. Additionally, even though the actual loss for training in this setting appears in the Appendix G, would good if there was a direct reference from the paper.*
>
> **A**:  Thank you very much for your suggestion. We added the description of RLHF and the object for $h_t^\theta$ in Section 3.5 in the revised version.
>
> **Q**: * In the main paper a single choice of probability path is introduced in equation 4. However going over the appendix it seems a different choice of probability path was used in the experiment, the metric induced path as in equation 21. Further, this probability path seems to use an additional metric which is not specified. While it is reasonable to leave these equations in the Appendix, it is expected to mention it in the experimental setup.*
>
> **A**:  Thank you very much for your suggestion. We added more description of the probability path of equation 21 into the experimental setup in the revised version. Specifically, we use the probability path of equation 4 in our simulations in Section 4.1. And we use the  probability path of 21 in Section 4.2 since it used in our baseline FUDOKI. The metric used in FUDOKI is $d(f(x),f(z))=\|\|\tilde{x}-\tilde{z}\|\|^4$, where $\tilde{x}, \tilde{z}$ are normalized token embeddings which are taken from the original text embedding layer of Janus-Pro-7B and the image embeddings of LlamaGen; for time schedule $\beta_t$, we set $\beta_t=3(t/(1-t))^{0.9}$ as suggested in [1]. We also include the details in the Appendix H.2.
>
> [1] Neta Shaul, Itai Gat, Marton Havasi, Daniel Severo, Anuroop Sriram, Peter Holderrieth, Brian Karrer, Yaron Lipman, and Ricky T. Q. Chen. Flow matching with general discrete paths: a kinetic-optimal perspective. ICLR, 2025
>
> **Q**: *Almost no implementation detail is provided for experiment: which pre-trained model is used is not clear, architecture and model size are not specified, and number of training iteration as well as effective batch size.*
>
> **A**:  Thank you very much for your reminder. For the experiments on multimodal benchmarks, we adopt the pre-trained FUDOKI from [2], which has model size 1.5M. Our guidance model is a network composed of 6 LLaDA blocks with hidden dimension 768, a time embedding mapping $\mathbb{R}$ to a hidden dimension of 768, a token embedding mapping each token into a 768-dimensional space. We include the details in Appendix H.2.
>
> [2] Wang, J., Lai, Y., Li, A., Zhang, S., Sun, J., Kang, N., Wu, C., Li, Z., & Luo, P. (2025). FUDOKI: Discrete Flow-based Unified Understanding and Generation via Kinetic-Optimal Velocities. ArXiv, abs/2505.20147.
>
> **Q**: *It might be standard approach but I still find it disturbing that the GenEval reward model was used both in training and evaluation.*
>
> **A**:  Thank you very much for your question. This setup follows the standard RLHF pipeline commonly adopted in prior work [3–6]. The key rationale is that, although the same reward model is used during both training and evaluation, the inputs are strictly separated: only the prompts/images from the training set are used for training, while prompts/images from the test set are reserved exclusively for evaluation.
>
> [3] Liu, J., Liu, G., Liang, J., Li, Y., Liu, J., Wang, X., Wan, P., Zhang, D., & Ouyang, W. (2025). Flow-GRPO: Training Flow Matching Models via Online RL. NeurIPS.
>
> [4] Xue, Z., Wu, J., Gao, Y., Kong, F., Zhu, L., Chen, M., Liu, Z., Liu, W., Guo, Q., Huang, W., & Luo, P. (2025). DanceGRPO: Unleashing GRPO on Visual Generation. ArXiv, abs/2505.07818.
>
> [5] He, X., Fu, S., Zhao, Y., Li, W., Yang, J., Yin, D., Rao, F., & Zhang, B. (2025). TempFlow-GRPO: When Timing Matters for GRPO in Flow Models. ArXiv, abs/2508.04324.
>
> [6] Li, J., Cui, Y., Huang, T., Ma, Y., Fan, C., Yang, M., & Zhong, Z. (2025). MixGRPO: Unlocking Flow-based GRPO Efficiency with Mixed ODE-SDE. ArXiv, abs/2507.21802.

---

> > ### Author Response · Authors · 2025-11-21
> >
> > **Q**: *The sampling Section is lacking, going over the appendix it seems there is a strong connection between previous suggested sampling algorithm and the exact guided sampling algorithm. Stressing the difference in the main paper could help the reader.*
> >
> > **A**:  Thank you very much for your question. The main difference between the guided sampling and previous suggested sampling algorithm is that in each sampling step we need to calculate the transition probability of target distribution $ q^ {(d,\theta)} _{1 \mid t}(\cdot \mid \mathbf{x} _t) $, which is a product of the guidance network and pretrained transition probability $ p^ {(d,\theta)} _{1 \mid t}(\cdot \mid \mathbf{x} _t) $. We stressed the difference in Section 3.4 in the revised version.
> >
> > **Q**: *I find at least two mathematical claims that where introduced in the main paper but were not actually used by the proposed methods (negative claims):
> > a. Theorem 2 is introduced and in the next three lines it is already stated that the authors do not make any use of it.
> > b. Remark 1, this remark seems to claim again that the direct approximation of the rate would not yield the desired flow.*
> >
> > **A**:  Thank you very much for your question. In this paper, we would like to provide a comprehensive framework for guidance of discrete flow matching. In Theorem 1, we provide the exact guidance of the transition probability. In Theorem 2, we provide the exact guidance of the transition rate matrix. Moreover, we have the following findings:
> > 1. We found that Equation 2 in existing work [7] is a special case of Theorem 2 when the target distribution is a conditional distribution of the source distribution.
> > 2. We found the first-order approximation used in [7,8,9] might not be appropriate for discrete space, we mention it in Remark 1.
> > 3. We found that the computational costs of applying guidance on the transition rate matrix are computationally costly than applying guidance on transition probability, which is summarized in Table 1 and also verified in Section 4.1.
> >
> >
> > [7] Nisonoff, H.M., Xiong, J., Allenspach, S., & Listgarten, J. (2025). Unlocking Guidance for Discrete State-Space Diffusion and Flow Models. ICLR.
> >
> > [8] Schiff, Y., Sahoo, S.S., Phung, H., Wang, G., Boshar, S., Dalla-torre, H., Almeida, B.P., Rush, A., Pierrot, T., & Kuleshov, V. (2025). Simple Guidance Mechanisms for Discrete Diffusion Models. ICLR.
> >
> > [9] Vignac, C., Krawczuk, I., Siraudin, A., Wang, B., Cevher, V., & Frossard, P. (2023). DiGress: Discrete Denoising diffusion for graph generation. ICLR.
> >
> >
> > **Q**: *In Algorithm 2 Sampling with Posterior-Based Guidance it is state that the guided posterior needs to be calculated at each step, Since the sequence length and vocabulary size can potentially be large, how does this affect the sampling cost compared to the vanilla sampling in Algorithm 1?*
> >
> > **A**:  Thank you very much for your question. At each step, we input the current state $\mathbf{x}_t$ into both the guidance network and pretrained posterior network, and then multiply the output to get the final transition probability in the target distribution $q^{d,\theta} _{1|t}(\cdot|\mathbf{x}_t)$ to update the current state $\mathbf{x}_t$ like Algorithm 1. Since the number of parameters in our guidance network is much smaller than that in the pretrained model in practice, so the extra computational cost is much smaller than the cost in Algorithm 1 which is used for sampling the source data distribution $p_1$.

---

### Meta-Review · Area_Chair_QZ9o · 2026-01-08

**Summary:**

This paper introduces a framework for guiding Discrete Flow Matching (DFM) models. Unlike prior methods that rely on first-order approximations (which can be inaccurate in discrete spaces), this work derives an "exact" guidance term for the transition rate/probability. The method is applied to conditional generation (inverse problems) and RLHF on text-to-image tasks.

The initial scores was moderately positive (scores 6, 4, 6), with reviewers appreciating the mathematical rigor of the exact derivation. The 4th review from reviewer nrvw has been flagged as problematic and an AI-generated review (rating 2) by AC, therefore it will be scrutinized.
Concerns were raised mainly regarding the clarity of the presentation (confusing terminology between "guidance" and "RL"), missing proofs (CTMC conservativeness), lack of implementation details, and the absence of high-dimensional benchmarks beyond the specific multimodal setup used.

**Reviewer Concerns:**

Reviewer 375z's mainly questions the presentations and some missing details, reviewer gFdm requests clarification of framing RL as guidance framework. The rebuttal is overall able to clarify major questions.

**Addressed**:
- Theoretical Soundness (N6g7): The reviewer questioned if the derived transition rates satisfy the conservation of probability ($\sum Q_{ij} = 0$). The authors provided a derivation in the rebuttal proving they do.
- Implementation Details (375z): Missing details about the pretrained model (FUDOKI), architecture, and probability path were added to the appendix.
- Terminology (gFdm): The reviewer criticized the use of "guidance" for a method that requires training (RL). The authors clarified that their framework unifies training-free (energy-guided) and training-based (RLHF) methods under one mathematical umbrella, justifying the terminology.

**Partially Addressed**:
- Experimental Scope (375z, N6g7): Reviewers noted the lack of diverse high-dimensional benchmarks (e.g., standard discrete image generation like ImageNet 32x32). The authors stuck to their multimodal (FUDOKI) setup and 2D toy examples. While valid, this limits the assessment of the method's generality across different discrete domains.
- Negative Density Ratios (N6g7): The reviewer worried about negative probabilities if density ratios aren't constrained. The authors claimed this doesn't happen because they use predefined ratios (classifiers/rewards) rather than estimating them.

**Reviewer Scores:**

- Reviewer 375z (Original: 6): Predicted: 6 (Marginally Above). The added details cleared up confusion, but the fundamental "heavy math" presentation and cost concerns likely prevent a higher score.

- Reviewer gFdm (Original: 4): Predicted: 6 (Marginally Above). The reviewer indicates low confidence on the theoretical expertise. The clarification on why RL fits their guidance framework addresses their main concern, and the theoretical soundness was validated by other reviewers.

- Reviewer N6g7 (Original: 6): Actual: 6 (Marginally Above). The reviewer explicitly stated in the discussion: "I have no further questions and maintain my positive rating." They accepted the theoretical proofs but didn't feel compelled to raise the score to an 8.

- Reviewer nrvw (Original 2): AI-generated low-quality review flagged. No rebuttal is provided, and the previous AC has taken action. I chose to downweight this review.

---

### Decision · Program_Chairs · 2026-01-26

Accept (Poster)